# Online Learning of Pure States is as Hard as Mixed States

**Maxime Meyer**
Department of Mathematics & IPAL, IRL2955
National University of Singapore
Singapore
maxime.meyer@u.nus.edu

**Soumik Adhikary**
Centre for Quantum Technologies
National University of Singapore
Singapore
soumik@nus.edu.sg

**Naixu Guo**
Centre for Quantum Technologies
National University of Singapore
Singapore
naixug@u.nus.edu

**Patrick Rebentrost**
Centre for Quantum Technologies & School of Computing
National University of Singapore
Singapore
cqtfpr@nus.edu.sg

## Abstract

Quantum state tomography, the task of learning an unknown quantum state, is a fundamental problem in quantum information. In standard settings, the complexity of this problem depends significantly on the type of quantum state that one is trying to learn, with pure states being substantially easier to learn than general mixed states. A natural question is whether this separation holds for any quantum state learning setting. In this work, we consider the online learning framework and prove the surprising result that learning pure states in this setting is as hard as learning mixed states. More specifically, we show that both classes share almost the same sequential fat-shattering dimension, leading to identical regret scaling. We also generalize previous results on full quantum state tomography in the online setting to (i) the $\epsilon$-realizable setting and (ii) learning the density matrix only partially, using *smoothed analysis*.

## 1 Introduction

Learning information from an unknown quantum state is a fundamental task in quantum physics. An $N$-dimensional quantum state $\rho$ is represented as an $N \times N$ positive semi-definite Hermitian matrix with unit trace. Given several perfect copies of $\rho$, full quantum state tomography seeks to reconstruct the complete matrix representation of $\rho$ via measurements. It has a wide range of practical applications, including tasks such as characterizing qubit states for superconducting circuits [Lucero et al., 2008], nitrogen-vacancy (NV) centers in diamond [Neumann et al., 2010], and verifying successful quantum teleportation [Bouwmeester et al., 1997]. The ease of learning a quantum state is often characterized by the sample complexity, *i.e.* the number of independent copies of the quantum state required for an accurate reconstruction. For a general state $\rho$, the sample complexity scales as $\tilde{\Theta}(N^3)$ for incoherent measurements. However, if the state is known to be pure, the sample complexity can be improved to $\tilde{\Theta}(N)$ [Kueng et al., 2017, Haah et al., 2016, Chen et al., 2023] . Nevertheless, for $n$-qubit states, where $N = 2^n$, this scaling implies an exponential dependence on the number of qubits.

However, for many practical problems, such as certification of a quantum device [Gross et al., 2010, Flammia and Liu, 2011, Eisert et al., 2020] and property estimation for quantum chemistry [Huang et al., 2021, Wu et al., 2023, Guo et al., 2024, Raza et al., 2024, Miller et al., 2024], only partial

39th Conference on Neural Information Processing Systems (NeurIPS 2025).

information about the quantum state is needed. To address such cases, quantum PAC learning, also known as pretty good tomography, was introduced by Aaronson [2007]. In this framework, a fixed distribution $\mathcal{D}$ governs the selection of two-outcome measurements $E$, which are represented by $N \times N$ Hermitian matrices with eigenvalues in $[0, 1]$. The learner then tries to output a hypothesis state $\omega$ such that $\mathrm{Tr}(E\rho) \underset{E \sim \mathcal{D}}{\approx} \mathrm{Tr}(E\omega)$ with high probability. The number of measurements required to output this hypothesis $n$-qubit state was shown to scale only linearly with the number of qubits $n$, yielding an exponential improvement over full-state tomography. However, a key limitation of both these approaches is that they do not account for adversarial environments, where the set of realizable measurements may evolve over time.

This limitation can be circumvented by generalizing to the online learning setting [Aaronson et al., 2018, Chen et al., 2024, Bansal et al., 2024], where learning a quantum state $\rho$ is posed as a $T$-round repeated two-player game. In each round $t \in [T]$, Nature–also called the adversary–chooses a measurement $E_t$ from the set of two-outcome measurements. The task of the learner is to predict the value $\mathrm{Tr}(E_t\rho)$ by selecting a hypothesis $\omega_t$ and computing $\mathrm{Tr}(E_t\omega_t)$ based on previous results. Thereafter, Nature returns the loss, a metric quantifying the difference between the prediction and the true value. The most adversarial scenario arises when the measurement at each round is chosen from the set of all two-outcome measurements without any constraints, *i.e.*, it can be chosen adversarially and adaptively. It has been shown that in such cases, a learner can output a hypothesis state which incurs an additional $O(\sqrt{nT})$ loss compared to the best possible hypothesis state after $T$ rounds of the game [Aaronson et al., 2018]. This measure of how much worse a learner performs compared to the best possible strategy in hindsight is called regret and serves as a fundamental measure of performance for any online learning problem.

Given the clear separation in sample complexity between pure and mixed state tomography [Kueng et al., 2017, Haah et al., 2016], we ask the central question that this work aims to address:

*Is there any separation between online learning of pure and mixed quantum states?*

In this work we show that the answer is **No**. Our proof is based on the analysis of the sequential fat-shattering dimension of pure and mixed states. Informally, it can be seen as the minimum number of mistakes–defined as errors exceeding a threshold $\delta$–that a learner must make before successfully learning a quantum state $\rho$ against a perfect adversary. We have shown that both pure and mixed states share almost the same sequential fat-shattering dimension of order $\Theta(\frac{\delta^2}{n})$, and hence the same regret. Indeed, it has been demonstrated that both upper and lower bounds on regret can be expressed in terms of this dimension [Rakhlin et al., 2015b].

We believe that this result is surprising. Indeed, not only is there a significant difference in sample complexities between learning pure and mixed states in the standard tomographic settings, but this distinction also holds in specific online learning scenarios. For instance, under bandit feedback with adaptive measurements, Lumbreras et al. [2022, 2024] shows that the regret for mixed states grows exponentially faster than for pure states.

A prior work [Aaronson et al., 2018] established tight bounds on the sequential fat-shattering dimension of general mixed states. In this work, we establish lower bounds on the sequential fat-shattering dimension – and consequently on the regret – of several subclasses of quantum states, the most important of which is the class of pure states. Our approach employs a distinct proof strategy, providing new insights and extending naturally to various subsettings of the online quantum state learning problem. As a result, we show that the regret for online learning of both pure and mixed quantum state scales as $\Theta(\sqrt{nT})$.

A key feature of the online learning setting considered here is the adversary's ability to select measurements in a completely unconstrained manner. As a result, this setting may effectively incur full state tomography, even in cases where only partial information about the state is needed. We therefore introduce the concept of smoothness for online learning of quantum states. Smoothed analysis was first introduced in Spielman and Teng [2004] as a tool that allows for interpolation between the worst and the average case analysis. Later, Haghtalab et al. [2024] extended this concept to the online setting, where the degree of adversariality is quantified by a smoothness parameter $\sigma \in [0, 1]$. The particular value $\sigma = 1$ corresponds to independent and identically distributed (i.i.d.) inputs, while the limit $\sigma \to 0$ corresponds to fully adversarial inputs. In this work, we establish an

upper bound on regret for smoothed online learning of quantum states, providing insights into the effect of adversariality on regret scaling.

## 1.1 Main Contributions

**Equivalence between pure and mixed state learning:** We show that pure and mixed states share almost the same sequential fat-shattering dimension, leading to identical regret scaling under the $L_1$ loss in Section 4. We also prove a novel dependence of minimax regret with $L_2$ loss on the Rademacher complexity. This allows us to extend the tightness of regret to the $L_2$ loss setting for both pure and mixed states.

**Extensions of Online Learning of Quantum States to more realistic settings:** We obtain new regret bounds for two settings of interest, which capture more accurately the settings encountered by experimentalists, in Section 5. The first is the $\epsilon$-realizable setting, where the learner is able to measure $\text{Tr}(E_t \rho)$ with error $\epsilon$ at every round. The second is the smooth setting, where the learner is only interested about learning specific properties of the quantum state $\rho$.

## 1.2 Related Works

**Pure and mixed state tomography:** In full state tomography of an $N$ dimensional quantum state $\rho$, the goal is to reconstruct its complete classical representation given several independent copies. The associated sample complexity refers to the number of copies required to obtain a classical description of $\rho$ up to an accuracy $\epsilon$. It has been shown that the sample complexity up to trace distance $\epsilon$ is $\tilde{\Theta}(Nr/\epsilon^2)$ for incoherent measurements [Haah et al., 2016, Kueng et al., 2017, Chen et al., 2023], where $r$ is the rank of the density matrix $\rho$. This result highlights a fundamental separation in the sample complexities between pure and mixed state tomography, as the rank of pure states is $r = 1$. Given the fundamental nature of this separation in quantum information science, one might expect it to persist in the online learning setting. In fact, Lumbreras et al. [2022] studies the online learning of properties of quantum states under bandit feedback with adaptive measurements, obtaining a regret scaling of $\Theta(\sqrt{T})$ for mixed states. Subsequently, Lumbreras et al. [2024] improved this result to $\Theta(\text{polylog } T)$ for pure states with rank-1 projective measurements, showing an exponential separation. In contrast, our results demonstrate that, in the general online learning setting, the regret scaling for pure and mixed states is identical.

**Existing bounds:** Aaronson et al. [2018] showed that the sequential fat-shattering dimension of quantum states with parameter $\delta$ is tight, of order $\Theta(\frac{n}{\delta^2})$. They also use a result from Arora et al. [2012] to show that the regret is tight for the $L_1$ loss in the non-realizable case (that is when the data isn't assumed to come from an actual quantum state), being of order $\Theta(\sqrt{nT})$. In this paper, we generalize this result to several new settings of interest. We also introduce a new proof technique that allows us to provide lower bounds for sequential fat-shattering dimension for restricted settings, notably proving near-tightness for pure states in Theorem 4.1.

## 2 Preliminaries

### 2.1 Quantum States and Measurements

A general mixed $n$-qubit (quantum bit) quantum state $\rho$ corresponds to a Hermitian positive semi-definite matrix of size $2^n$ and of trace 1. It will be called a **pure** state if and only if it is of rank 1 (or equivalently if and only if $\text{Tr}(\rho^2) = 1$). We can thus identify any pure state $\rho$ to a vector in the $2^n$ dimensional Hilbert space $\mathbb{C}^{2^n}$, typically expressed in the Dirac notation as $|\psi\rangle$. Under this notation, any pure state vector can be expressed as $|\psi\rangle = \sum_{i=1}^{2^n} c_i |i\rangle$ with $c_i \in \mathbb{C}$ and $\sum_{i=1}^{2^n} |c_i|^2 = 1$. Here, $\{|i\rangle\}_{i=1}^{2^n-1}$ corresponds to the canonical basis of the vector space. For a pure state $|\psi\rangle$ the corresponding density matrix representation is given by $\rho = |\psi\rangle\langle\psi|$, where $\langle\psi| = |\psi\rangle^\dagger$ is the complex conjugate of $|\psi\rangle$.

Information from quantum states can be obtained via quantum measurements. In our work we will focus on two outcome measurements. They are represented by two-element positive operator-valued measure (POVM) $\{E, \mathbf{1} - E\}$, where $E \in \text{Herm}_{\mathbb{C}}(2^n)$ and $\text{Spec}(E) \subset [0, 1]$. Since the second element of the POVM is uniquely determined by the first, a two-outcome measurement can

effectively be represented by a single operator $E$. A measurement $E$ is said to *accept* a quantum state $\rho$ with probability $\mathrm{Tr}(E\rho)$ and *reject* it with probability $1 - \mathrm{Tr}(E\rho)$. For a given quantum state $\rho$, predicting its acceptance probabilities for all measurements $E$ is tantamount to characterizing it completely. Hence learning a quantum state $\rho$ is equivalent to learning the function $\mathrm{Tr}_\rho$, defined as $\mathrm{Tr}_\rho(E) = \mathrm{Tr}(E\rho)$.

## 2.2 Online Learning of Quantum States

Online learning, or the sequential prediction model, is a $T$ round repeated two-player game [Cesa-Bianchi and Lugosi, 2006]. In each round $t \in [T]$ of the game, the learner is presented with an input from the sample space $x_t \in \mathcal{X}$. Without any loss of generality, we can assume that $x_t$ is sampled from a distribution $\mathcal{D}_t(\mathcal{X})$ where $\mathcal{D}_t$ may be chosen adversarially. The learner's goal is to learn an unknown function $h : \mathcal{X} \to \mathcal{Y}$ from the data they receive. Here, $\mathcal{Y}$ denotes the space of possible labels for each input $x_t$. The learning proceeds by designing an algorithm that outputs a sequence of functions $h_t : \mathcal{X} \to \mathcal{Y}$ chosen from a hypothesis class $\mathcal{H}$. After each round, the learner incurs a loss and aims to minimize the cumulative regret–which corresponds to the difference with the loss incurred by the best hypothesis in hindsight–at the end of all $T$ rounds of the game. The main figure of merit used in online learning, which represents the regret of the best strategy from the learner, when presented with the hardest possible inputs at every round is called the minimax regret [Rakhlin et al., 2015a].

**Definition 2.1** (Minimax regret). Let $\mathcal{X}$ denote the sample space, $\mathcal{Y}$ its associated label space, and $\mathcal{H}$ the hypothesis class. Let $\mathcal{P}$ and $\Delta(\mathcal{H})$ be sets of probability measures defined on $\mathcal{X}$ and $\mathcal{H}$ respectively. In each round $t \in [T]$ of the online learning process, the learner incurs a loss $\ell_t(h_t(x_t), y_t)$, where $y_t \in \mathcal{Y}$ is the true label associated to $x_t$. The minimax regret is then defined as:

$$\mathcal{V}_T = \Big\langle \inf_{\mathcal{Q}\in\Delta(\mathcal{H})} \sup_{y_t} \sup_{\mathcal{D}_t\in\mathcal{P}} \mathbb{E}_{h_t\sim\mathcal{Q}} \mathbb{E}_{x_t\sim\mathcal{D}_t} \Big\rangle_{t=1}^T \Big[ \sum_{t=1}^T \ell_t(h_t(x_t), y_t) - \inf_{h\in\mathcal{H}} \sum_{t=1}^T \ell_t(h(x_t), y_t) \Big], \quad (1)$$

where $\big\langle \,\cdot\, \big\rangle_{t=1}^T$ denotes iterated application of the enclosed operators.

The question of learnability of an online learning problem can then be reduced to the study of $\mathcal{V}_T$. Given a pair $(\mathcal{H}, \mathcal{X})$, a problem is said to be online learnable if and only if $\lim_{T\to\infty} \mathcal{V}_T/T = 0$.

In the context of quantum state learning, we define the sample space as $\mathcal{X} \subset \{E \in \mathrm{Herm}_\mathbb{C}(2^n), \mathrm{Spec}(E) \subset [0,1]\}$. Denoting $\mathcal{C}_n$ as the set of all $n$-qubit quantum states, we set the hypothesis class to be $\mathcal{H}_n = \{\mathrm{Tr}_\omega, \omega \in \mathcal{C}_n\}$, and $\mathcal{Q}$ can be seen as a distribution over $\mathcal{C}_n$. Here, the learner receives a sequence of measurements $(E_t)_{t\in[T]}$, each drawn from a distribution $\mathcal{D}_t$ (chosen adversarially) one at a time. Upon receiving each measurement, the learner selects a hypothesis $\omega_t \in \mathcal{C}_n$ and thereby incurs a loss of $\ell_t(\mathrm{Tr}_{\omega_t}(E_t), y_t) = \ell_t(\mathrm{Tr}(E_t\omega_t), y_t)$. In practice, for quantum tomography, $\ell_t$ is taken to be either the $L_1$ or the $L_2$ loss (that is $\ell_t(a, b) \in \{|a - b|, (a - b)^2\}$). Lastly, the label $y_t \in \mathcal{Y} = [0,1]$ is revealed to the learner. It may be an approximation of $\mathrm{Tr}(E_t\rho)$, but it is allowed to be arbitrary in general.

## 2.3 Sequential fat-shattering dimension

The main notion we will be studying in this paper is that of sequential fat-shattering dimension.

**Definition 2.2** (Sequential fat-shattering dimension). A $\mathcal{X}$-valued complete binary tree $\mathbf{x}$ of depth $T$ is deemed to be $\delta$-shattered by a hypothesis class $\mathcal{H}$ if there exists a real-valued complete binary tree $\mathbf{v}$ of same depth $T$ such that for all paths $\boldsymbol{\epsilon} \in \{\pm 1\}^{T-1}$,

$$\exists\, h \in \mathcal{H} \,:\, \forall t \in [T] \;\; \epsilon_t[h(\mathbf{x}_t(\boldsymbol{\epsilon})) - \mathbf{v}_t(\boldsymbol{\epsilon})] \geq \frac{\delta}{2}.$$

The sequential fat-shattering dimension at scale $\delta$, $\mathrm{sfat}_\delta(\mathcal{H}, \mathcal{X})$, is defined to be the largest $T$ for which $\mathcal{H}$ $\delta$-shatters a $\mathcal{X}$-valued tree of depth $T$.

Recall that a $\mathcal{X}$-valued complete binary tree of depth $T$, $\mathbf{x}$, is defined as a sequence of $T$ mappings $(\mathbf{x}_1, \mathbf{x}_2, \cdots, \mathbf{x}_T)$, where $\mathbf{x}_t : \{\pm 1\}^{t-1} \to \mathcal{X}$, with a constant function $\mathbf{x}_1 \in \mathcal{X}$ as the root. In simpler terms the tree can be seen as a collection of $T$ length paths $\boldsymbol{\epsilon} = (\epsilon_1, \epsilon_2, \cdots, \epsilon_{T-1}) \in \{\pm 1\}^{T-1}$ ($+1$

indicating right and $-1$ indicating left from any given node) and $\mathbf{x}_t(\boldsymbol{\epsilon}) \equiv \mathbf{x}_t(\epsilon_1, \cdots, \epsilon_{t-1}) \in \mathcal{X}$ denoting the label of the $t$-th node on the corresponding path $\boldsymbol{\epsilon}$.

This dimension is a fundamental property in online learning, as it both upper and lower bounds regret [Rakhlin et al., 2015a,b], as shown in Equations (2) and (3).

$$\mathcal{V}_T \le \inf_{\alpha > 0} \left\{ 4\alpha T L - 12L\sqrt{T} \int_\alpha^1 \sqrt{\mathrm{sfat}_\delta(\mathcal{H}, \mathcal{X}) \log\left(\frac{2eT}{\delta}\right)} d\delta \right\}. \tag{2}$$

Note that this upper bound also holds for $\bar{\mathcal{V}}_T$. The bound in Equation (2) was used in Aaronson et al. [2018] to derive the regret upper bounds for online quantum state learning. Similarly, the minimax regret can also be lower bounded by the sequential fat-shattering dimension, provided that $\ell_t(h_t(x_t), y_t) = |h_t(x_t) - y_t|$ and that $\mathcal{P}$ is taken to be the whole set of all distributions on $\mathcal{X}$.

$$\mathcal{V}_T \ge \frac{1}{4\sqrt{2}} \sup_{\delta > 0} \left\{ \sqrt{\delta^2 T \min\{\mathrm{sfat}_\delta(\mathcal{H}, \mathcal{X}), T\}} \right\}. \tag{3}$$

## 3 Lower Bounds for Online Learning of Quantum States

In this section, we employ a distinct proof strategy than Aaronson et al. [2018] to obtain lower bounds on sequential fat-shattering dimension–recall that they proved $\mathrm{sfat}_\delta(\mathcal{H}_n, \mathcal{X}) = \Theta(\frac{n}{\delta^2})$. This allows us to extend those bounds naturally to various subsettings of the online quantum state learning problem, characterized by a restricted hypothesis class $\mathcal{H} \subset \mathcal{H}_n$ and a constrained sample space $\mathcal{X}$. Such settings frequently arise in practical applications, where the focus is on characterizing specific subsets of quantum states. Furthermore, experimental constraints often limit the implementable set of measurement operations. We derive bounds on $\mathrm{sfat}(\mathcal{H}, \mathcal{X})$ for several such practically relevant subsettings, leading up to the most general formulation of the learning problem.

**Intuition of proofs:** Our proofs are based on the construction of $\mathcal{X} \times [0, 1]$-valued trees, which can be looked upon as a means to extract information about a quantum state $\rho$ at every layer. After a certain number of layers, full information about the state is recovered. The number of layers achieved serves as a lower bound to the sequential fat-shattering dimension. Therefore, the goal is to construct the largest tree before full information is recovered.

In our construction, we define gaining information simply as approximating the coefficients of the density matrix $\rho$. At each layer, we approximate a different coefficient as seen in Section 3.2. This is how we construct the $\mathcal{X}$-valued part of the tree. We can then approximate each coefficient to an error $\epsilon$ using the Halving Tree defined in Section 3.1 (Section 3.3 shows how to combine both trees). The intuition behind the choice of the coefficients we approximate comes from the study of chordal graphs in Section 3.4. An easier way to understand it is that we consider only the first row of the matrix, as it is of rank 1. Finally, we find the optimal error $\epsilon$ to obtain the lower bound.

To set the stage for the following sections, we first establish a few notations: since we will frequently consider pure states, the quantum state $\boldsymbol{\omega}(\boldsymbol{\epsilon})$ will be denoted by its associated vector $|\psi(\boldsymbol{\epsilon})\rangle$, where $\boldsymbol{\omega}(\boldsymbol{\epsilon}) = |\psi(\boldsymbol{\epsilon})\rangle \langle \psi(\boldsymbol{\epsilon})|$. Furthermore, we define $N = 2^n$ to be the dimension of the Hilbert space under consideration. In addition, we will denote the pair of binary trees $(\mathbf{x}, \mathbf{v})$ by a single tree $\mathbf{T}$. We call $\mathbf{x}$ the $\mathcal{X}$-valued part of $\mathbf{T}$, and $\mathbf{v}$ the real-valued part of $\mathbf{T}$.

**We highlight** that while many of the theorems in this section can be recovered from existing results, our contribution lies in introducing a unified proof framework. This scheme not only underpins our main result in Section 4, but also offers a versatile foundation that can be adapted to various sub-settings of interest.

### 3.1 Learning with respect to a single measurement

We start with the learning problem of estimating the expectation value of an unknown $n$-qubit quantum state with respect to a fixed measurement operator $E$. This learning problem is key to practical tasks such as quantum state discrimination and hypothesis testing [Barnett and Croke, 2009, Bae and Kwek, 2015]. Formally, we take $\mathcal{X} = \{E\}$ to be the sample space and keep $\mathcal{H}_n$ as the hypothesis class. As mentioned previously, we are focused on providing a lower bound on $\mathrm{sfat}_\delta(\mathcal{H}_n, \mathcal{X})$, which could then be used to lower bound $\mathcal{V}_T$. We achieve this by constructing what

we call the Halving tree $\mathbf{T}_h$, which has a constant $\mathcal{X}$-valued part, and a real-valued part as shown in Figure 1. Every halving tree $\mathbf{T}_h[i,T] = (\mathbf{x}, \mathbf{v})$ will thus be entirely determined by its depth $T$ and the constant measurement $|i\rangle\langle i|$. The name follows from the distinctive structure exhibited by the real part of the tree $\mathbf{T}_h[i,T]$, as shown in Figure 1. This construction will prove useful as a crucial building block for establishing the regret bounds in more general settings (see Theorems 3.4, 3.6 and 4.1).

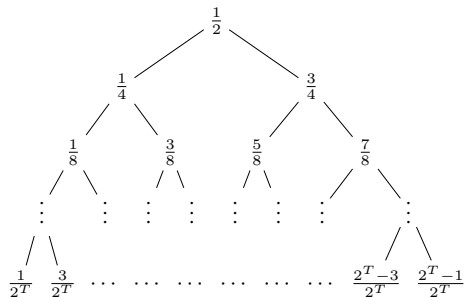

Figure 1: Real-valued part of the halving tree $\mathbf{T}_h$, up to $\frac{1}{N}$ factor.

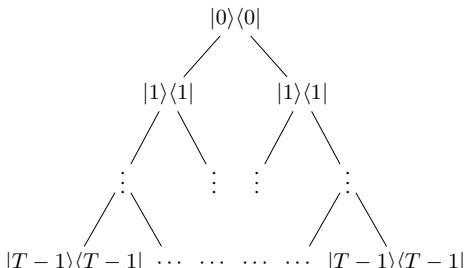

Figure 2: $\mathcal{X}$-valued part of the Von Neumann tree $\mathbf{T}_{vn}$.

**Theorem 3.1.** *Let $E \in \mathrm{Herm}_{\mathbb{C}}(2^n), \mathrm{Spec}(E) \subset [0,1]$ be a fixed measurement, and $\mathcal{X} = \{E\}$ be the sample space. Let $\mathcal{H}_n = \{\mathrm{Tr}_\omega, \omega \in \mathcal{C}_n\}$ be the hypothesis class, where $\mathcal{C}_n$ is the set of all $n$-qubit quantum states. Then we have $\mathrm{sfat}_\delta(\mathcal{H}_n, \mathcal{X}) = \Omega(\log_2(\frac{1}{\delta}))$.*

We prove this result in Appendix A.

*Remark* 3.2. Note that the lower bound on the fat-shattering dimension obtained above is independent of $n$, and therefore still holds if the hypothesis class is induced by 1-qubit pure states.

### 3.2 Learning uniform superposition states

We now shift our attention to a harder setting. Consider the sample space $\mathcal{X}$ consisting of the $N$ measurements corresponding to an orthogonal basis of $\mathcal{C}_n$. The hypothesis class will be induced by the uniform superpositions of basis states. Such states play an important role in fundamental quantum algorithms [Simon, 1994, Shor, 1997, Grover, 1997], and for quantum random number generators [Mannalatha et al., 2023]. We now provide a lower bound to the sequential fat-shattering dimension for this specific setting. We accomplish this by constructing what we call the Von Neumann tree $\mathbf{T}_{vn}$. The name follows from the fact that, while its real-valued part is constant, all nodes in the $\mathcal{X}$-valued part of $\mathbf{T}_{vn}$ are labelled by Von Neumann measurements as shown in Figure 2.

**Theorem 3.3.** *Let $(|0\rangle, ..., |N-1\rangle)$ be an orthogonal basis of $\mathcal{C}_n$, where $\mathcal{C}_n$ is the set of all $n$-qubit quantum states. Denote the sample space as $\mathcal{X} = \{|i\rangle\langle i|, i \in [\![0, N-1]\!]\}$. Let $\mathcal{H} = \{\mathrm{Tr}_\omega, \omega = \frac{1}{\sqrt{|I|}} \sum_{i \in I} |i\rangle, I \subset [\![0, N-1]\!]\}$ be the hypothesis class. Then we have $\mathrm{sfat}_\delta(\mathcal{H}, \mathcal{X}) = \Omega(\min(\frac{1}{\delta}, 2^n))$.*

We prove this result in Appendix B.

### 3.3 Learning general states using Von Neumann measurements

Building on the setting established in the previous section, we consider the sample space $\mathcal{X}$ consisting of the $N$ measurements corresponding to an orthogonal basis of $\mathcal{C}_n$. The hypothesis class in the present setting is however induced by the set of all pure quantum states. The corresponding learning problem involves a learner estimating the expectation values of an unknown $n$-qubit quantum state with respect to $N$ measurement operators, where the hypothesis is chosen from the set of all $n$-qubit pure quantum states. Related problems have been considered, for example, in Zhao et al. [2023].

We now provide a lower bound to the sequential fat-shattering dimension for this specific setting. We accomplish this by constructing what we call the Von Neumann Halving tree $\mathbf{T}_{vnh}$, which is constructed by combining $\mathbf{T}_h$ and $\mathbf{T}_{vn}$ as shown in Figure 3. Our construction illustrates the utility of $\mathbf{T}_h$, demonstrating its effectiveness as a tool to multiply existing lower bounds by a factor of $n$.

$$|0\rangle\langle 0| \; \frac{\mathbf{T}_h[0,T]}{\qquad\qquad} \; |1\rangle\langle 1| \; \frac{\mathbf{T}_h[1,T]}{\qquad\qquad} \; \cdots \; \frac{\mathbf{T}_h[N-3,T]}{\qquad\qquad} \; |N-2\rangle\langle N-2| \; \frac{\mathbf{T}_h[N-2,T]}{\qquad\qquad}$$

Figure 3: Von Neumann Halving Tree (Horizontal)

**Theorem 3.4.** *Let $(|0\rangle, ..., |N-1\rangle)$ be an orthogonal basis of $\mathcal{C}_n$ and $\mathcal{X} = \{|i\rangle\langle i|, i \in [\![0, N-1]\!]\}$ be the sample space. Let $\mathcal{H} = \{\mathrm{Tr}_\omega, \omega \in \mathcal{C}_n, \mathrm{Tr}(\omega^2) = 1\}$ be the hypothesis class. Here $\mathcal{C}_n$ is the set of all $n$-qubit quantum states. Then, for $\delta = 2^{-\frac{n}{\eta}}$, we have $\mathrm{sfat}_\delta(\mathcal{H}_n, \mathcal{X}) = \Omega(\frac{n}{\delta^\eta}) \, \forall \eta < 1$.*

We prove this result in Appendix C.

### 3.4 Tightness of sequential fat-shattering dimension of quantum states

Having considered several restricted settings in the previous sections, we now turn to the most general formulation of the online quantum state learning problem. Formally, we define the sample space to be the space of all 2-outcome measurements $\mathcal{X}$, while the Hypothesis class is given as $\mathcal{H}_n$. Our objective is to derive a tight lower bound on $\mathrm{sfat}_\delta(\mathcal{H}_n, \mathcal{X})$. To accomplish this, we will use the techniques developed in the previous sections. In addition, we will establish new results on completion of partial matrices, which are essential for our analysis. We start with the latter.

Let us first introduce a few necessary definitions on partial matrices and their completions. A partial matrix is a matrix in which certain entries are specified while the other entries are free to be chosen. It is called partial symmetric if it is symmetric on the specified entries. And a completion of a partial matrix refers to a specific assignment of values to its unspecified entries. The main result we will use is the following, proved in Appendix D.

**Lemma 3.5.** *Any real partial symmetric matrix $\omega$ satisfying the following conditions*

1. *$w_{11} = \frac{1}{2}$, and $w_{ii} = \frac{1}{2(N-1)} \; \forall i \in [\![2, N]\!]$,*

2. *Elements are specified on the set $\{w_{1i}, w_{i1}\}$, where $i \in [N]$,*

3. *$\forall i \in [\![2, N]\!], |w_{1i}| \leq \frac{1}{2\sqrt{N-1}}$,*

*can be completed to a density matrix. We will denote $\mathrm{part}(w_{12}, w_{13}, \cdots, w_{1N})$ such a matrix.*

We now derive the lower bound for $\mathrm{sfat}_\delta(\mathcal{H}_n, \mathcal{X})$. The key idea is to construct a new tree analogous to the Von Neumann halving tree, with a crucial distinction that it accommodates more general measurements, extending beyond the $|i\rangle\langle i|$ type measurements that have been considered thus far. Furthermore, given this tree, we apply Lemma 3.5 to ensure that all paths on the tree can be associated to a valid density matrix. We show that this allows us to obtain the quadratic dependence on $\frac{1}{\delta}$ in the lower bound of $\mathrm{sfat}_\delta(\mathcal{H}_n, \mathcal{X})$, thus establishing the almost tightness.

**Theorem 3.6.** *Let $\mathcal{X} = \{E \in \mathrm{Herm}_\mathbb{C}(2^n), \mathrm{Spec}(E) \subset [0,1]\}$ be the sample space. Define $\mathcal{H}_n = \{\mathrm{Tr}_\omega, \omega \in \mathcal{C}_n\}$ as the hypothesis class, where $\mathcal{C}_n$ is the set of all $n$-qubit quantum states. Then, for $\delta = 2^{-\frac{n}{\eta}}$, we have $\mathrm{sfat}_\delta(\mathcal{H}_n, \mathcal{X}) = \Omega(\frac{n}{\delta^\eta}), \forall \eta < 2$.*

We prove this result in Appendix E. Note that this result directly implies tightness of regret for $L_1$ loss [Aaronson et al., 2018], aswell as for $L_2$ loss as shown in Appendix G.

*Remark* 3.7. As mentioned in the beginning of this section, our method for lower bounding the sequential fat-shattering dimension differs from that employed in Aaronson [2007], and is adaptable to various restricted online quantum state learning settings. For cases involving further restrictions, whether on the sample space, the hypothesis class or the relationship between $\delta$ and $n$ (Recall that Aaronson [2007] required $\delta \geq \sqrt{n2^{-(n-5)/35}/8}$) we conjecture that the techniques from Appendix E involving matrix completion would serve as a valuable foundation.

## 4 Online Learning of Pure States is as Hard as Mixed States

While the tightness of the sequential fat-shattering dimension, and hence of the minimax regret were already known, we emphasize that the results were derived for a hypothesis class being induced by

the set of all $n$-qubit quantum states. In this section, we consider a more restrictive setting where the hypothesis class is induced solely by the set of all $n$-qubit pure states. The derivation closely follows the techniques developed in the previous section with a notable distinction: whereas the previous sections relied on matrix completion results (without defining the states $\boldsymbol{\omega}(\boldsymbol{\epsilon})$ explicitly), here we shall construct the states $\boldsymbol{\omega}(\boldsymbol{\epsilon}) = |\psi(\boldsymbol{\epsilon})\rangle\langle\psi(\boldsymbol{\epsilon})|$ explicitly. We show that the bounds on the sequential fat-shattering dimension and consequently the minimax regret remain tight in this setting.

**Theorem 4.1.** *Let $\mathcal{X} = \{E \in \mathrm{Herm}_{\mathbb{C}}(2^n), \mathrm{Spec}(E) \subset [0,1]\}$ be the sample space. Define $\mathcal{H} = \{\mathrm{Tr}_\omega, \omega \in \mathcal{C}_n, \mathrm{Tr}[\omega^2] = 1\}$ as the hypothesis class, where $\mathcal{C}_n$ is the set of all $n$-qubit quantum states. Then, for $\delta = 2^{-\frac{n}{\eta}}$, we have $\mathrm{sfat}_\delta(\mathcal{H}, \mathcal{X}) = \Omega(\frac{n}{\delta\eta}), \forall \eta < 2$.*

Now, building on the result in Theorem 4.1, we proceed to demonstrate the tightness of the minimax regret $\mathcal{V}_T$, proving that it is asymptotically the same as for mixed states:

**Corollary 4.2.** *Let $\mathcal{X} = \{E \in \mathrm{Herm}_{\mathbb{C}}(2^n), \mathrm{Spec}(E) \subset [0,1]\}$ be the sample space. Define $\mathcal{H} = \{\mathrm{Tr}_\omega, \omega \in \mathcal{C}_n, \mathrm{Tr}[\omega^2] = 1\}$ as the hypothesis class, where $\mathcal{C}_n$ is the set of all $n$-qubit quantum states. Then we have $\mathcal{V}_T = \Omega(\sqrt{nT})$, assuming the loss function under consideration is the $L_1$-loss. This result still holds for the $L_2$ loss, as long as $T \leq 4^n$.*

We prove these results in Appendices F and G respectively.

An important aspect to notice here is that the significance of the sequential fat-shattering dimension extends beyond regret analysis. Several corollaries follow from Theorem 4.1. One such example is the fact that Theorem 1 from Aaronson et al. [2018], which has been shown to be optimal for mixed states, is also optimal for pure states.

**Corollary 4.3.** *Let $\rho$ be an $n$-qubit mixed state, and let $E_1, E_2, \ldots$ be a sequence of 2-outcome measurements that are revealed to the learner one by one, each followed by a value $b_t \in [0,1]$ such that $|\mathrm{Tr}(E_t\rho) - b_t| \leq \varepsilon/3$. Then there is an explicit strategy for outputting hypothesis states $\omega_1, \omega_2, \ldots$ such that $|\mathrm{Tr}(E_t\omega_t) - \mathrm{Tr}(E_t\rho)| > \varepsilon$ for at most $O\left(\frac{n}{\varepsilon^2}\right)$ values of $t$. This mistake bound is almost asymptotically optimal, even if we restrict $\rho$ to be a pure state.*

## 5 Extensions of Online Learning of Quantum States

### 5.1 The $\epsilon$-realizable setting

In practice, an experimenter does not have access to the exact value of $\mathrm{Tr}(E_t\rho)$. Assuming completely adversarial feedback is overly pessimistic and renders the learning problem particularly challenging. This motivates the consideration of the $\epsilon$-*realizable setting*, where the learner incurs the $L_1$ loss with respect to a fixed quantum state $\rho$, and receives noisy feedback $y_t$ at each round. Specifically, the feedback $y_t$ is an approximation of $\mathrm{Tr}(E_t\rho)$ with error parametrized by $\epsilon$. For instance, $y_t$ may be drawn from a uniform distribution over the interval $[\mathrm{Tr}(E_t\rho) - \epsilon, \mathrm{Tr}(E_t\rho) + \epsilon]$, or from a Gaussian distribution $\mathcal{N}(\mathrm{Tr}(E_t\rho), \epsilon)$. More generally, any non-deterministic distribution $\mu$ centered at $\mathrm{Tr}(E_t\rho)$ is permissible.

We show that even in this more favorable setting, the regret admits the same lower bound as in the fully adversarial case–*asymptotically independent of $\epsilon$.*

$$\bar{\mathcal{V}}_T = \left\langle \inf_{\mathcal{Q}\in\Delta(\mathcal{C}_n)} \sup_{\mathcal{D}_t\in\mathcal{P}} \mathbb{E}_{\omega_t\sim\mathcal{Q}} \mathbb{E}_{E_t\sim\mathcal{D}_t} \mathbb{E}_{y_t\sim\mu} \right\rangle_{t=1}^T \sup_{\rho\in\mathcal{C}_n} \left[ \sum_{t=1}^T |\mathrm{Tr}(E_t\omega_t) - \mathrm{Tr}(E_t\rho)| \right]. \quad (4)$$

**Theorem 5.1.** *The minimax regret in the $\epsilon$-realizable setting satisfies $\bar{\mathcal{V}}_T = \Theta(\sqrt{nT})$. In particular, these bounds are independent of $\epsilon$.*

We prove this result in Appendix H.

### 5.2 Smoothed Online Learning of Quantum States

The online learning framework discussed thus far is fully adversarial, allowing the adversary to select arbitrary two-outcome measurements at each round $t$. However, as highlighted in Section 1, practical

learning tasks often target specific properties of the quantum state $\rho$, rather than aiming for full state tomography. Within the PAC learning paradigm, such tasks are typically modelled by fixing a distribution $\mathcal{D}$ over the space of measurements, reflecting the learner's interest in a particular subset of observables.

To bridge the adversarial and distributional regimes, we adopt the lens of *smoothed analysis*, which interpolates between worst-case and average-case models. Specifically, we require that at each round $t$, the adversary selects a distribution $\mathcal{D}_t$ over measurements that remains close to the target distribution $\mathcal{D}$. This models an adversary with limited power to perturb the measurement process.

This smoothed perspective is not merely a theoretical convenience — it captures an important aspect of experimental practice. Indeed, quantum devices rarely implement measurements with perfect fidelity. Instead, small temporal fluctuations in control parameters, thermal drift, or calibration errors can cause the realized measurement to deviate slightly from the intended one Leibfried et al. [2003]. From a learning-theoretic standpoint, these deviations can be modelled as stochastic perturbations: at each round, the performed measurement is sampled from a distribution that is $\varepsilon$-close to the nominal one. Thus, smoothed analysis offers a principled framework for reasoning about online learning in the presence of structured noise in the measurement process.

**Definition 5.2** (Smooth distributions). A distribution $\mu$ is said to be $\sigma$-smooth with respect to a fixed distribution $\mathcal{D}$ for a $\sigma \in (0, 1]$ if and only if [Haghtalab et al., 2020]:

1. $\mu$ is absolutely continuous with respect to $\mathcal{D}$, *i.e.* every measurable set $A$ such that $\mathcal{D}(A) = 0$ satisfies $\mu(A) = 0$

2. The Radon Nikodym derivative $d\mu/d\mathcal{D}$ satisfies the following relation:

$$\operatorname{ess\,sup} \frac{d\mu}{d\mathcal{D}} \leq \frac{1}{\sigma}. \tag{5}$$

Let $\mathcal{B}(\sigma, \mathcal{D})$ be the set of all $\sigma$-smooth distributions with respect to $\mathcal{D}$. In smoothed online learning, the adversary is restricted by the condition $\mathcal{D}_t \in \mathcal{B}(\sigma, \mathcal{D})$. Note that in this setting we recover the case of an oblivious adversary for $\sigma = 1$, while we get the completely adversarial case for $\sigma \to 0$.

Recall the expression of minimax regret in Equation (1). In the smoothed setting, the expression gets slightly modified accounting for the restriction imposed on the adversary:

$$\mathcal{V}_T = \Big\langle \inf_{\mathcal{Q} \in \Delta(\mathcal{H})} \sup_{\mathcal{D}_t \in \mathcal{B}(\sigma, \mathcal{D})} \mathbb{E}_{h_t \sim \mathcal{Q}} \mathbb{E}_{x_t \sim \mathcal{D}_t} \Big\rangle_{t=1}^T \Big[ \sum_{t=1}^T \ell_t(h_t(x_t)) - \inf_{h \in \mathcal{H}} \sum_{t=1}^T \ell_t(h(x_t)) \Big]. \tag{6}$$

Here, the key difference with Equation (1) is that $\mathcal{D}_t$ is now restricted to the set of all $\sigma$-smooth distributions with respect to $\mathcal{D}$ instead of all possible distributions on $\mathcal{X}$. We recall that for quantum state learning, we have $\mathcal{X} \subset \{E \in \operatorname{Herm}_{\mathbb{C}}(2^n), \operatorname{Spec}(E) \subset [0, 1]\}$, $\mathcal{H}_n = \{\operatorname{Tr}_\omega, \omega \in \mathcal{C}_n\}$ and a target state $\rho$. For the sake of brevity, we will continue using the notation $x_t$ to indicate input data and $h$ to indicate the hypothesis. The derivation here closely follows the approach in Block et al. [2022] (which derives the regret bounds for classical smoothed online supervised learning) with one important difference; in the original derivation, for a given input $x_t \in \mathcal{X}$, the authors distinguish between a predicted label $\hat{y}_t \in \mathcal{Y}$ and $h_t(x_t) \in \mathcal{Y}$ where $\mathcal{Y}$ is the label space. We do not make this distinction as our labels are always related to our inputs via the hypothesis.

**Theorem 5.3.** *Let* $\mathcal{X} = \{E \in \operatorname{Herm}_{\mathbb{C}}(2^n), \operatorname{Spec}(E) \subset [0, 1]\}$ *be the sample space. Define the hypothesis class* $\mathcal{H} = \{\operatorname{Tr}_\omega, \omega \in \mathcal{C}_n, \operatorname{Tr}[\omega^2] = 1\}$ *as the set of n-qubit pure states, where* $\mathcal{C}_n$ *is the set of all n-qubit quantum states. Furthermore let* $\sigma \in (0, 1]$ *be the smoothness parameter. Then we have* $\mathcal{V}_T = O\left( \sqrt{\frac{nT \log T}{\sigma}} \right).$

We prove this result in Appendix K.

## 6   Conclusion and Limitations

**Lower bounds on fat-shattering dimension:** In this work, we established lower bounds on sequential fat-shattering dimension of various subproblems within the quantum state learning framework.

Crucially, we showed that pure and mixed states almost share the same asymptotical dimension. Note that, although our construction directly implies that the regular $\delta$-fat-shattering dimension of pure states scales as $\Omega(\frac{1}{\delta^2})$, whether we can recover $\Omega(\frac{n}{\delta^2})$ for pure $n$-qubits in the offline setting remains an open question.

**Consequences on regret:** This lower bound on sequential fat-shattering dimension has several implications, including the key result that learning pure and mixed states in the online setting will incur the same asymptotical regret for the $L_1$-loss. However, the lower bound for the $L_2$ loss might leave room for improvement. Additionally, sequential fat-shattering dimension may serve as a fundamental tool for deriving bounds on other key complexity measures in quantum state learning.

**Smoothed online learning:** Finally, we extend our analysis from standard online learning of quantum states to the smoothed online learning setting. To our knowledge, this work represents the first application of smoothed analysis to quantum state learning. In this setting, we establish an upper bound on the regret. However a key open question is whether this bound is tight.

**Possible extensions to classical learning theory:** While a general (mixed) quantum state can be represented as a positive semi-definite Hermitian matrix of rank (up to) $2^n$, a pure state can be represented as a positive semi-definite Hermitian matrix of rank 1. The fact that this restriction on the rank of a matrix has no impact on the sequential fat-shattering dimension lower bound (and hence the regret lower bound) is highly non-intuitive and does not seem to apply in varied settings of classical learning: Alon et al. [2016] shows that rank of sign matrices impact VC dimension, Bartlett et al. [2022] shows that the $\epsilon$-fat shattering dimension of learned SCW matrices of rank $k$ is $O\left(n \cdot \left(m + k \log\left(\frac{n}{k}\right) + \log\left(\frac{1}{\epsilon}\right)\right)\right)$, and Srebro and Shraibman [2005] shows that the pseudodimension of matrices scale linearly with the rank.

The reason why this scaling of dimension with rank vanishes in quantum is the additional assumptions on the density matrices, which are semi-definite positive, Hermitian, and of trace 1. Such matrices have however been extensively studied in Learning Theory [Tsuda et al., 2004, Shen et al., 2008, 2009]. In addition, common matrices satisfying those conditions are normalized covariance and kernel matrices, and Laplacian matrices differ only for the fact that their trace is constant equal to $2m$, where $m$ is the number of edges of the associated graph. We leave as future work to see how our results translate to these other settings of interest.

## Acknowledgements

The work of Maxime Meyer, Soumik Adhikary, Naixu Guo, and Patrick Rebentrost was supported by the National Research Foundation, Singapore and A*STAR under its CQT Bridging Grant, its AI Singapore Programme (AISG Award No: AISG3-PhD-2026-01-068T) and its Quantum Engineering Programme under grant NRF2021-QEP2-02-P05. Authors thank Josep Lumbreras, Aadil Oufkir, and Jonathan Allcock for their insightful discussions and kind suggestions.

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

# A Proof of Theorem 3.1

*Proof.* Without loss of generality, we set $\mathcal{X} = \{|i\rangle\langle i|\}$, with $i \in [\![0, N-2]\!]$. Define $\mathbf{x}$ as the complete binary tree of depth $T$ such that $\forall t \in [T], \forall \boldsymbol{\epsilon} \in \{\pm 1\}^{T-1}$,

$$\mathbf{x}_t(\boldsymbol{\epsilon}) = |i\rangle\langle i|. \tag{7}$$

Furthermore, define $\mathbf{v}$ (Figure 1) as the complete binary tree of depth $T$ such that $\forall t \in [T], \forall \boldsymbol{\epsilon} \in \{\pm 1\}^{T-1}$,

$$\mathbf{v}_t(\boldsymbol{\epsilon}) = \frac{1}{2^t}\sum_{k=0}^{t-1}\epsilon_k 2^{t-k-1} = \sum_{k=0}^{t-1}\epsilon_k 2^{-k-1}. \tag{8}$$

Here, we set $\epsilon_0 = 1$.

Given $(\mathbf{x}, \mathbf{v})$, we now set:

$$|\psi(\boldsymbol{\epsilon})\rangle = \sqrt{\sum_{k=0}^{T-1}\epsilon_k 2^{-k-1}}|i\rangle + \sqrt{1 - \sum_{k=0}^{T-1}\epsilon_k 2^{-k-1}}|\perp\rangle. \tag{9}$$

Then, for $T = \lfloor \log_2(\frac{1}{\delta}) \rfloor$ (which implies $\delta \leq \frac{1}{2^T}$), $\forall \boldsymbol{\epsilon} \in \{\pm 1\}^{T-1}, \forall t \in [T]$, we have:

$$
\begin{aligned}
\epsilon_t[\mathrm{Tr}_{\boldsymbol{\omega}(\boldsymbol{\epsilon})}(\mathbf{x}_t(\boldsymbol{\epsilon})) - \mathbf{v}_t(\boldsymbol{\epsilon})] &= \epsilon_t\left[\sum_{k=0}^{T-1}\epsilon_k 2^{-k-1} - \sum_{k=0}^{t-1}\epsilon_k 2^{-k-1}\right] \\
&= \epsilon_t\sum_{k=t}^{T-1}\epsilon_k 2^{-k-1} \\
&= 2^{-t-1} + \epsilon_t\sum_{k=t+1}^{T-1}\epsilon_k 2^{-k-1} \\
&\geq 2^{-t-1} - \sum_{k=t+1}^{T-1}2^{-k-1} \\
&= 2^{-t-1} - (2^{-t-1} - 2^{-T}) \\
&\geq \delta,
\end{aligned}
\tag{10}
$$

where the first inequality follows from the minimum value that the term $\epsilon_t\sum_{k=t+1}^{T-1}\epsilon_k 2^{-k-1}$ can take, and the last inequality is by direct computation and the assumption $T = \lfloor\log_2(\frac{1}{\delta})\rfloor$. Thus, we show that the set $\mathcal{X}$ is $\delta$-shattered by the hypothesis class $\mathcal{H}_n$ with $\mathrm{sfat}_\delta(\mathcal{H}_n, \mathcal{X}) = \Omega(\log_2(\frac{1}{\delta}))$. $\square$

We can replace $\mathbf{v}$ by a slightly modified version of itself, where each node is scaled by a factor $\frac{1}{N}$. Therefore, the quantum state associated to a path has an amplitude corresponding to $|i\rangle$: $\mathrm{Tr}(|i\rangle\langle i|\boldsymbol{\omega}(\boldsymbol{\epsilon}))$ bounded by $\frac{1}{N}$. We will write $\mathbf{T}_h[i, T] = (\mathbf{x}, \mathbf{v})$ the resulting pair-valued tree and call it the Halving Tree of depth $T$. The index $i$ indicates the Von Neumann measurement associated to $\mathbf{x}$.

# B Proof of Theorem 3.3

*Proof.* Let $T \in [N]$. Define $\mathbf{v}$ as the complete binary tree of depth $T$ such that $\forall t \in [T], \forall \boldsymbol{\epsilon} \in \{\pm 1\}^{T-1}$,

$$\mathbf{v}_t(\boldsymbol{\epsilon}) = \frac{1}{2T}. \tag{11}$$

Furthermore, denote $\mathbf{x}$ (Figure 2) the complete binary tree of depth $T$ such that $\forall t \in [T], \forall \boldsymbol{\epsilon} \in \{\pm 1\}^{T-1}$,

$$\mathbf{x}_t(\boldsymbol{\epsilon}) = |t-1\rangle\langle t-1|. \tag{12}$$

We refer to the pair $(\mathbf{x}, \mathbf{v})$ as the Von-Neumann tree $\mathbf{T}_{vn}$.

Given the Von Neumann tree, we now associate each path $\epsilon$ to a pure quantum state:

$$|\psi(\epsilon)\rangle = \frac{1}{\sqrt{K+1}}\sum_{i=0}^{T-2}\mathbf{1}_{\epsilon_{i+1}=1}|i\rangle + \frac{1}{\sqrt{K+1}}|N-1\rangle, \tag{13}$$

where $K = \sum_{i=0}^{T-2}\mathbf{1}_{\epsilon_{i+1}=1}$, and $\mathbf{1}_{\epsilon=1}$ is an indicator function which takes value $1$ if $\epsilon = 1$ and $0$ otherwise. Then, for $\delta \le \frac{1}{2T}, \forall \, \epsilon \in \{\pm 1\}^{T-1}, \forall t \in [T]$:

$$\epsilon_t[\mathrm{Tr}_{\boldsymbol{\omega}(\epsilon)}(\mathbf{x}_t(\epsilon)) - \mathbf{v}_t(\epsilon)] = \epsilon_t\left[\frac{\mathbf{1}_{\epsilon_t=1}}{K+1} - \frac{1}{2T}\right]$$
$$\ge \delta. \tag{14}$$

Thus, we show that the set $\mathcal{X}$ is $\delta$-shattered by the hypothesis class $\mathcal{H}$ with $\mathrm{sfat}_\delta(\mathcal{H}, \mathcal{X}) = \Omega(\min(\frac{1}{\delta}, 2^n))$.

$\square$

## C  Proof of Theorem 3.4

*Proof.* Without any loss of generality, we can chose the sample space to be a set of Von Neumann measurements $\mathcal{X} = \{|i\rangle\langle i| : i \in [\![0, N-1]\!]\}$. Denote $t' = \lfloor\frac{t-1}{T}\rfloor$ and $\tilde{t} = t-1-Tt'$. We then define $\mathbf{x}$ as the complete binary tree of depth $T(N-1)$ such that $\forall t \in [T(N-1)], \forall \, \epsilon \in \{\pm 1\}^{T(N-1)-1}$,

$$\mathbf{x}_t(\epsilon) = |t'\rangle\langle t'|. \tag{15}$$

Furthermore, denote $\mathbf{v}$ the complete binary tree of depth $T(N-1)$ such that $\forall t \in [T(N-1)], \forall \, \epsilon \in \{\pm 1\}^{T(N-1)-1}$,

$$\mathbf{v}_t(\epsilon) = \frac{1}{2^{n+1}}(1 + \sum_{k=1}^{\tilde{t}}\epsilon_{k+Tt'}2^{-k}) \tag{16}$$

We refer to the pair $(\mathbf{x}, \mathbf{v})$ as the Von Neumann halving tree $\mathbf{T}_{vnh}$. The name follows from the fact that the trees $\mathbf{x}$ and $\mathbf{v}$ as defined in Equations (15) and (16) can be constructed by replacing each node on the $t$-th layer of $\mathbf{T}_{vn}$, for both $\mathcal{X}$ valued and real valued parts, by the corresponding parts of the halving tree $\mathbf{T}_h[t, T]$.

Given the Von Neumann halving tree we can now associate each path $\epsilon$ to a pure quantum state:

$$|\psi(\epsilon)\rangle = \sum_{i=0}^{N-2}\sqrt{a_i}|i\rangle + \sqrt{1 - \sum_{i=0}^{N-2}a_i}|N-1\rangle \tag{17}$$

where $\forall i \in [\![0, N-2]\!]$,

$$a_i = \mathbf{v}_{(i+1)T}(\epsilon) + \frac{1}{2^{T+n+1}}\epsilon_{(i+1)T}. \tag{18}$$

Then, for $\delta \le \frac{1}{2^{n+T+1}}$, we can show that $\forall \, \epsilon \in \{\pm 1\}^{T(N-1)-1}, \forall t \in [T(N-1)]$:

$$\epsilon_t[\mathrm{Tr}_{\boldsymbol{\omega}(\epsilon)}(\mathbf{x}_t(\epsilon)) - \mathbf{v}_t(\epsilon)] = \epsilon_t[\mathbf{v}_{(t'+1)T}(\epsilon) + \frac{1}{2^{T+n+1}}\epsilon_{(t'+1)T} - \frac{1}{2^{n+1}}(1 + \sum_{k=1}^{\tilde{t}}\epsilon_{k+Tt'}2^{-k})]$$

$$= \epsilon_t[\frac{1}{2^{n+1}}(1 + \sum_{k=1}^{T}\epsilon_{k+Tt'}2^{-k}) - \frac{1}{2^{n+1}}(1 + \sum_{k=1}^{\tilde{t}}\epsilon_{k+Tt'}2^{-k})]$$

$$= \frac{\epsilon_t}{2^{n+1}}(\sum_{k=\tilde{t}+1}^{T}\epsilon_{k+Tt'}2^{-k})$$

$$= \frac{1}{2^{n+1}}(\frac{1}{2^{\tilde{t}+1}} + \sum_{k=\tilde{t}+2}^{T}\epsilon_{k+Tt'}2^{-k})$$

$$\ge \delta \tag{19}$$

In particular, let $k \in \mathbb{N}^*$. Taking $\delta = \frac{1}{N^{1+\frac{1}{k}}}$ and $T = \lfloor \log_2(\frac{1}{4\delta N}) \rfloor$, we get $TN = \Omega(n\delta^{-\frac{1}{1+\frac{1}{k}}})$. Therefore, we have shown that the set $\mathcal{X}$ is $\delta$-shattered by the Hypothesis class $\mathcal{H}_n$ with $\mathrm{sfat}_\delta(\mathcal{H}_n) = \Omega(\frac{n}{\delta^\eta}), \forall \eta < 1$. $\qquad\square$

## D  Proof of Lemma 3.5

To prove Lemma 3.5, we rely on Theorem 7 in Grone et al. [1984]. We begin by introducing the necessary definitions. Let $\mathcal{G} = (V, E)$ be a finite undirected graph. A *cycle* in $\mathcal{G}$ is a sequence of distinct vertices $v_1, v_2, \ldots, v_s \in V$ such that $\{v_i, v_{i+1}\} \in E$ for all $i \in [s-1]$, and $\{v_s, v_1\} \in E$. A cycle is said to be *minimal* if and only if it has no chord, where a chord is an edge $\{v_i, v_j\} \in E$ with $|i - j| > 1$ and $\{i, j\} \neq \{1, s\}$.

A matrix $\omega$ is said to be $\mathcal{G}$-partial when its entries $w_{ij}$ are determined if and only if $\{i, j\} \in E$, while other elements are undetermined. A $\mathcal{G}$-partial matrix $\omega$ is said to be non-negative if and only if (a) $\omega_{ij} = \overline{\omega}_{ji}, \forall\{i, j\} \in E$ and (b) for any clique $\mathcal{C}$ of $\mathcal{G}$, the principal submatrix of $\omega$ corresponding to $\mathcal{C}$ (which has entries corresponding to $\mathcal{C}$) is positive semidefinite. Recall that a clique $\mathcal{C}$ of $\mathcal{G}$ is a complete subgraph of $\mathcal{G}$. The corresponding principal submatrix is obtained by keeping only the indices in $\mathcal{C}$.

A *completion* of a $\mathcal{G}$-partial matrix $\omega$ is a full Hermitian matrix $M$ such that $M_{ij} = \omega_{ij}$ for all $\{i, j\} \in E$. We say that M is a non-negative completion if and only if M is also positive semidefinite. A graph $\mathcal{G}$ is said to be *completable* if and only if any $\mathcal{G}$-partial non-negative matrix has a non-negative completion. With these definitions in place, we proceed to state the relevant results in Grone et al. [1984].

**Lemma D.1** (Grone et al. [1984]). *A graph $\mathcal{G}$ is completable if and only if every minimal cycle in the graph is of length $< 4$.*

Now that we have covered the necessary background, we can proceed to prove Lemma 3.5.

*Proof.* Let $\omega$ be a partial matrix as stated in Lemma 3.5. Consider the graph $\mathcal{G} = (V, E)$, where $V = [N]$ and $E = \{\{i, j\}, i \in [N], j \in \{1, i\}\} \cup \{\{i, i\}, i \in [N]\}$.

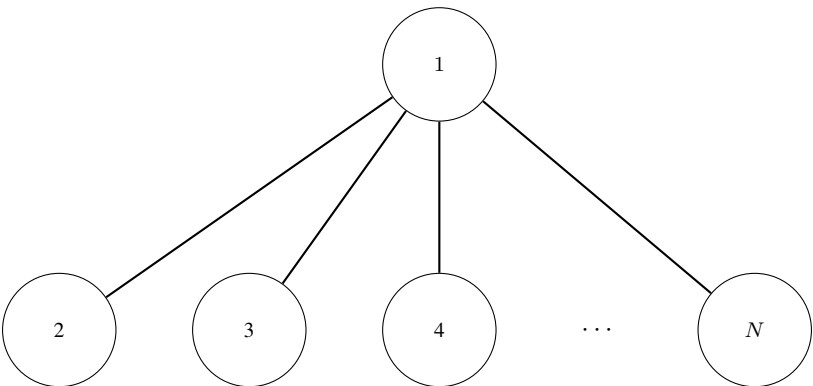

Figure 4: Graph representation of $\mathcal{G} = (V, E)$

One can easily check that $\mathcal{G}$ is completable using Lemma D.1. Therefore, if we prove that $\omega$ is non-negative $\mathcal{G}$-partial, it will have a positive semidefinite Hermitian completion. Combining with the first two conditions in Lemma 3.5, we can show that the completion of $\omega$ is Hermitian, positive semidefinite, with trace equal to 1, and therefore is a density matrix. Now to show that $\omega$ is non-negative $\mathcal{G}$-partial, let $\mathcal{C} = \{1, i\}$ be a clique of $\mathcal{G}$. Since $|w_{1i}| \leq \frac{1}{2\sqrt{N-1}}$ by assumption, the principal submatrix of $\omega$ corresponding to $\mathcal{C}$, i.e.,

$$\begin{pmatrix} \frac{1}{2} & \omega_{1i} \\ \omega_{i1} & \frac{1}{2(N-1)} \end{pmatrix} \tag{20}$$

can be shown to be non-negative. $\qquad\square$

$$\begin{bmatrix} \frac{1}{2} & w_{12} & \cdots & \cdots & w_{1N} \\ w_{12} & \frac{1}{2(N-1)} & ? & \cdots & ? \\ \vdots & ? & \frac{1}{2(N-1)} & \cdots & ? \\ \vdots & \vdots & \vdots & \ddots & \vdots \\ w_{1N} & ? & ? & \cdots & \frac{1}{2(N-1)} \end{bmatrix}$$

Figure 5: General form of the partial matrix described in Lemma 3.5. The interrogation marks denote the unspecified entries in the matrix.

## E    Proof of Theorem 3.6

*Proof.* Set $\mathcal{X} = \{E_{0,i}, i \in [N-1]\}$ where $E_{0,i} = \frac{1}{2}(|0\rangle\langle0| + |i\rangle\langle i| + |0\rangle\langle i| + |i\rangle\langle 0|)$. Define $t' = \lfloor \frac{t-1}{T} \rfloor$ and $\tilde{t} = t - 1 - Tt'$ for $T > 0$. Write $\mathbf{x}$ the complete binary tree of depth $T(N-1)$ such that $\forall t \in [T(N-1)], \forall \boldsymbol{\epsilon} \in \{\pm 1\}^{T-1}$,

$$\mathbf{x}_t(\boldsymbol{\epsilon}) = E_{0,t'+1}, \tag{21}$$

and $\mathbf{v}$ the complete binary tree of depth $T(N-1)$ such that $\forall t \in [T(N-1)], \forall \boldsymbol{\epsilon} \in \{\pm 1\}^{T-1}$,

$$\mathbf{v}_t(\boldsymbol{\epsilon}) = \frac{1}{4\sqrt{N-1}}(1 + \sum_{k=1}^{\tilde{t}} \epsilon_{k+Tt'} 2^{-k}). \tag{22}$$

The pair $(\mathbf{x}, \mathbf{v})$ resembles the Von Neumann halving tree constructed in the previous section, with the key difference being that the nodes in the $\mathcal{X}$ valued part of the new tree are now labelled by a different class of measurement operators. We will replace $\mathbf{v}$ with a slightly modified version $\tilde{\mathbf{v}}$:

$$\tilde{\mathbf{v}}_t(\boldsymbol{\epsilon}) = \mathbf{v}_t(\boldsymbol{\epsilon}) + \frac{1}{4}(1 + \frac{1}{N-1}). \tag{23}$$

We now associate every path $\boldsymbol{\epsilon}$ to any nonnegative completion $\boldsymbol{\omega}(\boldsymbol{\epsilon})$ of part$(a_1,...a_{N-1})$, where $\forall i \in [\![1, N-1]\!]$,

$$a_i = \mathbf{v}_{iT}(\boldsymbol{\epsilon}) + \frac{1}{2^{T+2}\sqrt{N-1}}\epsilon_{(i+1)T}. \tag{24}$$

Recall that the partial matrix part$(a_1,...a_{N-1})$ satisfies all the conditions in Lemma 3.5 and hence can be completed to a valid density matrix $\boldsymbol{\omega}(\boldsymbol{\epsilon})$.

Then, for $\delta \leq \frac{1}{2^{T+2}\sqrt{N-1}}$, we have that $\forall \boldsymbol{\epsilon} \in \{\pm 1\}^{T(N-1)-1}, \forall t \in [T(N-1)]$,

$$\epsilon_t[\text{Tr}_{\boldsymbol{\omega}(\boldsymbol{\epsilon})}(\mathbf{x}_t(\boldsymbol{\epsilon})) - \tilde{\mathbf{v}}_t(\boldsymbol{\epsilon})]$$

$$=\epsilon_t[\mathbf{v}_{(t'+1)T}(\boldsymbol{\epsilon}) + \frac{1}{2^{T+2}\sqrt{N-1}}\epsilon_{(t'+2)T} - \frac{1}{4\sqrt{N-1}}(1 + \sum_{k=1}^{\tilde{t}} \epsilon_{k+Tt'} 2^{-k})]$$

$$=\epsilon_t[\frac{1}{4\sqrt{N-1}}(1 + \sum_{k=1}^{T} \epsilon_{k+Tt'} 2^{-k}) - \frac{1}{4\sqrt{N-1}}(1 + \sum_{k=1}^{\tilde{t}} \epsilon_{k+Tt'} 2^{-k})]$$

$$=\frac{\epsilon_t}{4\sqrt{N-1}}(\sum_{k=\tilde{t}+1}^{T} \epsilon_{k+Tt'} 2^{-k})$$

$$=\frac{1}{4\sqrt{N-1}}(\frac{1}{2^{\tilde{t}+1}} + \sum_{k=\tilde{t}+2}^{T} \epsilon_{k+Tt'} 2^{-k})$$

$$\geq \delta. \tag{25}$$

By taking $\delta = \frac{1}{N^{\frac{1}{2}+\frac{1}{k}}}$ and $T = \lfloor \log_2(\frac{1}{4\delta\sqrt{N}}) \rfloor$, we get $TN = \Omega(n\delta^{-\frac{2}{1+\frac{2}{k}}})$. Therefore, we have shown that the set $\mathcal{X}$ is $\delta$-shattered by the Hypothesis class $\mathcal{H}_n$ with sfat$_\delta(\mathcal{H}_n) = \Omega(\frac{n}{\delta^\eta}), \forall \eta < 2$. $\square$

# F  Proof of Theorem 4.1

*Proof.* Set $\mathcal{X} = \{E_{0,i}, i \in [N-1]\}$ where $E_{0,i} = \frac{1}{2}(|0\rangle\langle 0| + |i\rangle\langle i| + |0\rangle\langle i| + |i\rangle\langle 0|)$. Define $t' = \lfloor \frac{t-1}{T} \rfloor$ and $\tilde{t} = t - 1 - Tt'$ for $T > 0$. Write $\mathbf{x}$ the complete binary tree of depth $T(N-1)$ such that $\forall t \in [T(N-1)], \forall \boldsymbol{\epsilon} \in \{\pm 1\}^{T-1}$,

$$\mathbf{x}_t(\boldsymbol{\epsilon}) = E_{0,t'+1}. \tag{26}$$

and $\mathbf{v}$ the complete binary tree of depth $T$ such that $\forall t \in [T], \forall \boldsymbol{\epsilon} \in \{\pm 1\}^{T-1}$,

$$\mathbf{v}_t(\boldsymbol{\epsilon}) = \frac{1}{4\sqrt{N-1}}(1 + \sum_{k=1}^{\tilde{t}} \epsilon_{k+Tt'} 2^{-k}). \tag{27}$$

We can associate every path $\boldsymbol{\epsilon}$ to a pure state

$$|\psi(\boldsymbol{\epsilon})\rangle = \frac{1}{\sqrt{2}}|0\rangle + \sum_{i=1}^{N-2} a_i |i\rangle + \left(\frac{1}{2} - \sum_{i=1}^{N-2} a_i^2\right)|N-1\rangle \tag{28}$$

where,

$$a_i = \mathbf{v}_{iT}(\boldsymbol{\epsilon}) + \frac{1}{2^{T+2}\sqrt{N-1}}\epsilon_{(i+1)T}. \tag{29}$$

Here $i \in [\![1, N-2]\!]$. Let $\mathbf{w}_t = \frac{1}{\sqrt{2}}\mathbf{v}_t + \frac{1}{2}(\frac{1}{2} + a_{t'+1}^2)$. Then, for $\delta \leq \frac{1}{2^{T+2}\sqrt{2(N-1)}}$, we have that $\forall \boldsymbol{\epsilon} \in \{\pm 1\}^{T(N-2)-1}, \forall t \in [T(N-2)]$,

$$\epsilon_t[\mathrm{Tr}_{|\psi(\boldsymbol{\epsilon})\rangle}(\mathbf{x}_t(\boldsymbol{\epsilon})) - \mathbf{w}_t(\boldsymbol{\epsilon})] = \frac{\epsilon_t}{\sqrt{2}}[\mathbf{v}_{(t'+1)T}(\boldsymbol{\epsilon}) + \frac{1}{2^{T+2}\sqrt{N-1}}\epsilon_{(t'+2)T} - \frac{1}{4\sqrt{N-1}}(1 + \sum_{k=1}^{\tilde{t}} \epsilon_{k+Tt'} 2^{-k})]$$

$$= \frac{\epsilon_t}{\sqrt{2}}[\frac{1}{4\sqrt{N-1}}(1 + \sum_{k=1}^{T} \epsilon_{k+Tt'} 2^{-k}) - \frac{1}{4\sqrt{N-1}}(1 + \sum_{k=1}^{\tilde{t}} \epsilon_{k+Tt'} 2^{-k})]$$

$$= \frac{\epsilon_t}{4\sqrt{2(N-1)}}(\sum_{k=\tilde{t}+1}^{T} \epsilon_{k+Tt'} 2^{-k})$$

$$= \frac{1}{4\sqrt{2(N-1)}}(\frac{1}{2^{\tilde{t}+1}} + \sum_{k=\tilde{t}+2}^{T} \epsilon_{k+Tt'} 2^{-k})$$

$$\geq \delta \tag{30}$$

In particular, let $k \in \mathbb{N}^*$. By taking $\delta = \frac{1}{N^{\frac{1}{2}+\frac{1}{k}}}$ and $T = \lfloor \log_2(\frac{1}{4\delta\sqrt{N}}) \rfloor$, we get $TN = \Omega(n\delta^{-\frac{2}{1+\frac{2}{k}}})$. Therefore, we have shown that the set $\mathcal{X}$ is $\delta$-shattered by the Hypothesis class $\mathcal{H}$ with $\mathrm{sfat}_\delta(\mathcal{H}, \mathcal{X}) = \Omega(\frac{n}{\delta^\eta}) \, \forall \eta < 2$. $\qquad \square$

# G  Proof of Corollary 4.2

*Proof for the $L_1$ loss.* From Rakhlin et al. [2015a,b], the minimax regret is lower bound by the sequential fat-shattering dimension as:

$$\mathcal{V}_T \geq \frac{1}{4\sqrt{2}} \sup_{\delta > 0} \left\{ \sqrt{\delta^2 T \min\{\mathrm{sfat}_\delta(\mathcal{H}_n, \mathcal{X}), T\}} \right\}$$

provided that the loss function under consideration is the $L_1$-loss. Now combining this result with the lower bound on $\mathrm{sfat}_\delta(\mathcal{H}_n, \mathcal{X})$ established in Theorem 3.6 we get $\mathcal{V}_T = \Omega(\sqrt{nT})$. $\qquad \square$

*Proof for the $L_2$ loss.* From Equation (1), and taking the $y_t$ to be Rademacher random variables, we get:

$$\mathcal{V}_T \geq \left\langle \sup_{x_t} \inf_{h_t \in \mathcal{H}} \mathbb{E}_{y_t} \right\rangle_{t=1}^T \left[ \sum_{t=1}^T (h_t(x_t) - y_t)^2 - \inf_{h \in \mathcal{H}} \sum_{t=1}^T (h(x_t) - y_t)^2 \right]$$

$$\geq \left\langle \sup_{x_t} \right\rangle_{t=1}^T \left[ \inf_{h_t \in \mathcal{H}} \sum_{t=1}^T (1 + h_t^2) + \mathbb{E}_{y_t} \sup_{h \in \mathcal{H}} \sum_{t=1}^T -(h(x_t) - y_t)^2 \right]$$

$$\geq \left\langle \sup_{x_t} \right\rangle_{t=1}^T \left[ \sum_{t=1}^T 1 + \mathbb{E}_{y_t} \sup_{h \in \mathcal{H}} \sum_{t=1}^T -(h(x_t) - y_t)^2 \right]$$

$$= \left\langle \sup_{x_t} \right\rangle_{t=1}^T \left[ \mathbb{E}_{y_t} \sup_{h \in \mathcal{H}} \sum_{t=1}^T 1 - (h(x_t) - y_t)^2 \right]$$

$$= \left\langle \sup_{x_t} \right\rangle_{t=1}^T \left[ \mathbb{E}_{y_t} \sup_{h \in \mathcal{H}} \sum_{t=1}^T 2 y_t h(x_t) - h^2(x_t) \right]$$

$$= \left\langle \sup_{x_t} \right\rangle_{t=1}^T \frac{1}{2} \left[ \sup_{h \in \mathcal{H}} \sum_{t=1}^T 2 h(x_t) - h^2(x_t) + \sup_{h \in \mathcal{H}} \sum_{t=1}^T -2 h(x_t) - h^2(x_t) \right]$$

$$\geq \left\langle \sup_{x_t} \right\rangle_{t=1}^T \frac{1}{2} \left[ \sup_{h \in \mathcal{H}} \sum_{t=1}^T h(x_t)(2 - h(x_t)) + (\sum_{t=1}^T -2 h(x_t) - h^2(x_t))_{h = \text{Tr} \frac{1}{N} I_N} \right]$$

$$\geq \left\langle \sup_{x_t} \right\rangle_{t=1}^T \frac{1}{2} \left[ \sup_{h \in \mathcal{H}} \sum_{t=1}^T h(x_t) + \sum_{t=1}^T -\frac{2}{N} - \frac{1}{N^2} \right]$$

$$\geq \mathcal{R}_T(\mathcal{H}) - \frac{3T}{2N} \tag{31}$$

Where we used the fact that

$$\mathcal{R}_T(\mathcal{H}) = \left\langle \sup_{x_t} \right\rangle_{t=1}^T \mathcal{R}_T(x, \mathcal{H})$$

$$= \left\langle \sup_{x_t} \right\rangle_{t=1}^T \mathbb{E}_y \left[ \sup_{h \in \mathcal{H}} \sum_{t=1}^T y_t h_t(x_t) \right]$$

$$= \left\langle \sup_{x_t} \right\rangle_{t=1}^T \frac{1}{2} \left[ \sup_{h \in \mathcal{H}} \sum_{t=1}^T h_t(x_t) - \sup_{h \in \mathcal{H}} \sum_{t=1}^T h_t(x_t) \right]$$

$$\leq \left\langle \sup_{x_t} \right\rangle_{t=1}^T \frac{1}{2} \left[ \sup_{h \in \mathcal{H}} \sum_{t=1}^T h_t(x_t) \right] \tag{32}$$

$\square$

## H  Proof of Theorem 5.1

*Proof.* All we need to do in order to lower bound $\bar{\mathcal{V}}_T$ is to exhibit a particular strategy $(\mathcal{D}_t)_{t \in [T]}$ the adversary can adopt. We will then show that such a strategy guarantees a minimax regret greater then $\Omega(\sqrt{nT})$, independently of the strategy picked by the learner.

It was shown in Aaronson [2007] that $n$-qubit quantum states, considered as a hypothesis class, have $(\delta, \frac{\delta^2}{n})$-fine-shattering dimension greater than $D = \lfloor \frac{n}{5\delta^2} \rfloor$. Let $T \in \mathbb{N}^*$ and write $\delta = \sqrt{\frac{n}{5T}}$. This means that there exists $(E_t)_{t \in [D]} \in \mathcal{X}^D$ such that $\forall (\epsilon_t) \in \{\pm 1\}^D, \exists \rho \in \mathcal{C}_n, \forall t \in [D]$:

$$\mathrm{Tr}(E_t\rho) \in \begin{cases} \left[\frac{1}{2} - \delta - \frac{\delta^2}{n}, \ \frac{1}{2} - \delta\right] & \text{if } \epsilon_t = -1, \\[2ex] \left[\frac{1}{2} + \delta, \ \frac{1}{2} + \delta + \frac{\delta^2}{n}\right] & \text{if } \epsilon_t = 1. \end{cases} \tag{33}$$

We will denote by $\boldsymbol{\rho} : 2^D \longrightarrow \mathcal{C}_n$ the function that associates every path to the corresponding quantum state in Equation (33). We can now show that if the adversary picks $E_t$ at every round $t$, no matter the strategy chosen by the learner, there exists a quantum state $\rho$ that will guarantee a loss greater than $\sqrt{nT}$.

For every $t \in [D]$, write $\epsilon_t = \mathrm{sgn}\left[\mathbb{E}_{\omega \sim \mathcal{Q}_t}(\frac{1}{2} - \mathrm{Tr}(E_t\omega))\right]$. Then, we clearly have that

$$\Big\langle \underset{\omega_t \sim \mathcal{Q}}{\mathbb{E}} \Big\rangle_{t=1}^{T} \Big[\sum_{t=1}^{T} |\,\mathrm{Tr}(E_t\omega_t) - \mathrm{Tr}(E_t\rho)|\Big] \geq D\delta$$

$$\geq \delta(T-1)$$

$$\geq \sqrt{\frac{nT}{5}} - 1 \tag{34}$$

*Remark* H.1. The last thing we need to check is that the feedback received by the learner doesn't give *much more* information than the interval of size $\frac{\delta^2}{n}$ in which lies $\mathrm{Tr}(E_t\rho)$, as defined in Equation (33). If $y_t \sim \mathcal{U}([\mathrm{Tr}(E_t\rho) - \epsilon, \mathrm{Tr}(E_t\rho) + \epsilon])$ for instance, this amounts to $\epsilon \geq \frac{\delta^2}{n}$, *id est* $T \geq \frac{1}{5\epsilon}$. For a general distribution, picking $T$ big enough such that the probability of $y_t \in [\mathrm{Tr}(E_t\rho) - \epsilon, \mathrm{Tr}(E_t\rho) + \epsilon]$ is greater than $\frac{1}{2}$ will only divide the lower bound by 2.

Finally, the fact that $\mathbb{E}(y_t) = \mathrm{Tr}(E_t\rho)$ allows us to conclude.

Therefore, $\bar{\mathcal{V}}_T = \Omega(\sqrt{nT})$. It is easy to see that $\bar{\mathcal{V}}_T \leq \mathcal{V}_T = O(\sqrt{nT})$. $\qquad\qquad\square$

# I   Regret bounds with sequential complexities in classical online learning

Notions of complexity for a given hypothesis class have traditionally been studied within the batch learning framework and are often characterized by the Rademacher complexity [Bartlett and Mendelson, 2003].

**Definition I.1** (Rademacher complexity). Let $\mathcal{X}$ be a sample space with an associated distribution $\mathcal{D}$ and $\mathcal{H}$ be the hypothesis class. Let $(x_j)_{j \in [m]} \sim \mathcal{D}^m$ be a sequence of samples, sampled i.i.d. from $\mathcal{X}$. The Rademacher complexity can then be defined as:

$$\mathcal{R}_m(\mathcal{H}) = \underset{(x_j) \sim \mathcal{D}^m}{\mathbb{E}} \Big[\frac{1}{m} \underset{\epsilon}{\mathbb{E}} \Big[\sup_{h \in \mathcal{H}} \sum_{j=1}^{m} \epsilon_j h(x_j)\Big]\Big],$$

where $\boldsymbol{\epsilon} = (\epsilon_1, \cdots, \epsilon_m)$ are called Rademacher variables, that satisfy $P(\epsilon = +1) = P(\epsilon = -1) = 1/2$.

Perhaps not surprisingly, in addition to being an indicator for expressivity of a given hypothesis class, Rademacher complexity also upper bounds generalization error in the setting of batch learning [Bartlett and Mendelson, 2003].

Rademacher complexity generalises to sequential Rademacher complexity in the online setting [Rakhlin et al., 2015a]. In order to define sequential Rademacher complexity let us first define a $\mathcal{X}$-valued complete binary tree.

**Definition I.2** (Sequential Rademacher complexity). Let $\mathbf{x}$ be a $\mathcal{X}$-valued complete binary tree of depth $T$. The sequential Rademacher complexity of a hypothesis class $\mathcal{H}$ on the tree $\mathbf{x}$ is then given as:

$$\mathfrak{R}_T(\mathcal{H}, \mathbf{x}) = \Big[\frac{1}{T} \underset{\boldsymbol{\epsilon}}{\mathbb{E}} \Big[\sup_{h \in \mathcal{H}} \sum_{t=1}^{T} \epsilon_t h(\mathbf{x}_t(\boldsymbol{\epsilon}))\Big]\Big].$$

The $\mathbf{x}$ dependence of the sequential Rademacher complexity can be subsequently removed by considering the supremum over all $\mathcal{X}$-valued trees of depth $T$: $\mathfrak{R}_T(\mathcal{H}) = \sup_{\mathbf{x}} \mathfrak{R}_T(\mathcal{H}, \mathbf{x})$. Similar to how Rademacher complexity upper bounds the generalization error, the sequential Rademacher complexity was shown to upper bound the minimax regret [Rakhlin et al., 2015a]. For the case of supervised learning, the following relation holds:

$$\mathcal{V}_T \leq 2LT\mathfrak{R}_T(\mathcal{H}). \tag{35}$$

Here, $L$ comes from the fact that the loss function considered is $L$-Lipschitz.

The growth of sequential Rademacher complexities has been shown to be influenced by other related notions of sequential complexities. One prominent example is the sequential fat-shattering dimension [Rakhlin et al., 2015a]. It was shown in Rakhlin et al. [2015a] that this dimension serves as an upper bound to the sequential Rademacher complexity, which subsequently provides a bound on the minimax regret as per Equation (35). Similarly, recall that the minimax regret can be lower bounded by the sequential fat-shattering dimension (Equation (3)), provided that $\ell_t(h_t(x_t), y_t) = |h_t(x_t) - y_t|$ and that $\mathcal{P}$ is taken to be the whole set of all distributions on $\mathcal{X}$. In fact, it is also lower bounded by the Rademacher complexity [Rakhlin et al., 2015a,b].

$$
\begin{aligned}
\mathcal{V}_T &\geq \left\langle \sup_{\mathcal{D}_t \in \mathcal{P}} \mathbb{E}_{x_t \sim \mathcal{D}_t} \right\rangle_{t=1}^T \mathbb{E}_{\boldsymbol{\epsilon}} \left[ \sup_{h \in \mathcal{H}} \sum_{t=1}^T \epsilon_t h(\mathbf{x}_t(\boldsymbol{\epsilon})) \right] \\
&\geq \frac{1}{4\sqrt{2}} \sup_{\delta > 0} \left\{ \sqrt{\delta^2 T \min\{\mathrm{sfat}_\delta(\mathcal{H}, \mathcal{X}), T\}} \right\}.
\end{aligned} \tag{36}
$$

where $\boldsymbol{\epsilon} = (\epsilon_1, \epsilon_2, \cdots, \epsilon_T)$ are Rademacher variables.

## J Coupling

To establish regret bounds in smoothed online learning, Haghtalab et al. [2024], Block et al. [2022] introduced the concept of coupling. The key idea here is that if the distributions $(\mathcal{D}_t)_{t=1}^T$ are $\sigma$-smooth with respect to $\mathcal{D}$, we may pretend that in expectation the data is sampled i.i.d from $\mathcal{D}$ instead of $(\mathcal{D}_t)_{t=1}^T$. For a more formal description, define $\mathcal{B}_T(\sigma, \mathcal{D})$ to be the set of joint distributions $\mathcal{D}_\wedge$ on $\mathcal{X}^T$, where each marginal distribution $\mathcal{D}_t(\cdot | x_1, ..., x_{t-1})$ is conditioned on the previous draws.

**Definition J.1** (Coupling). A distribution $\mathcal{D}_\wedge \in \mathcal{B}_T(\sigma, \mathcal{D})$ is said to be coupled to independent random variables drawn according to $\mathcal{D}$ if there exists a probability measure $\Pi$ with random variables $(x_t, Z_t^j)_{t \in [T], j \in [k]} \sim \Pi$ satisfying the following conditions:

1. $x_t \sim \mathcal{D}_t(\cdot | x_1.x_2, \cdots, x_{t-1})$,

2. $\{Z_t^j\}_{t \in [T], j \in [k]} \sim \mathcal{D}^{\otimes kT}$,

3. With probability at least $1 - Te^{-\sigma k}$, we have $x_t \in \{Z_t^j\}_{j \in [k]} \; \forall t \in [T]$.

The last relation is particularly interesting and is used to derive the regret bounds for smoothed online quantum state learning.

## K Proof of Theorem 5.3

For this proof, we will use the notion of Rademacher complexity and a few related results that can be found in Appendix I. We also use the concept of coupling described in Appendix J.

Consider the sequential Rademacher complexity $\mathfrak{R}_T(\ell \circ \mathcal{H}, \mathbf{x})$ in Definition I.2 defined on the function class $\ell \circ \mathcal{H}$. For the purpose of this proof let us consider a slightly modified version of the sequential Rademacher complexity defined as:

$$
\begin{aligned}
\mathfrak{R}_T(\ell \circ \mathcal{H}, \mathcal{D}_\wedge) &= \mathbb{E}_{\mathbf{x} \sim \mathcal{D}_\wedge} \mathfrak{R}_T(\ell \circ \mathcal{H}, \mathbf{x}) \\
&= \mathbb{E}_{\mathbf{x} \sim \mathcal{D}_\wedge} \left[ \frac{1}{T} \mathbb{E}_{\boldsymbol{\epsilon}} \left[ \sup_{h \in \mathcal{H}} \sum_{j=1}^T \epsilon_j \ell(h(\mathbf{x}_t(\boldsymbol{\epsilon})), h_\rho(\mathbf{x}_t(\boldsymbol{\epsilon}))) \right] \right],
\end{aligned} \tag{37}
$$

where $\mathcal{D}_\wedge \in \mathcal{B}_T(\sigma, \mathcal{D})$. Moreover we will call $\mathfrak{R}_T(\ell \circ \mathcal{H}, \mathcal{B}_T) = \sup_{\mathcal{D}_\wedge \in \mathcal{B}_T(\sigma, \mathcal{D})} \mathfrak{R}_T(\ell \circ \mathcal{H}, \mathcal{D}_\wedge)$. The key idea now is to relate $\mathfrak{R}_T(\ell \circ \mathcal{H}, \mathcal{D}_\wedge)$ to $\mathcal{R}(\mathcal{H})$ (Rademacher complexity assuming i.i.d. data inputs; see Definition I.1). This can be achieved using the idea of coupling discussed in the previous section.

**Lemma K.1.** *Let $\mathcal{D}_\wedge \in \mathcal{B}_T(\sigma, \mathcal{D})$ be a distribution that is coupled to independent random variables drawn according to the distribution $\mathcal{D}$, as per Definition J.1. Let $\mathcal{H}$ be a Hypothesis class and $\ell$ be the loss function. Then we have $\mathfrak{R}_T(\ell \circ \mathcal{H}, \mathcal{D}_\wedge) \leq T^2 e^{-\sigma k} + \mathcal{R}_{kT}(\ell \circ \mathcal{H})$*

*Proof.* Let $A$ be an event that $\mathbf{x}_t(\boldsymbol{\epsilon}) \in \{Z_t^j\}_{j=1}^k \ \forall t \in [T]$. Furthermore, let $\chi_A$ be the corresponding indicator function and $\chi_{A^c}$ be $1 - \chi_A$. Then we have:

$$\mathfrak{R}_T(\ell \circ \mathcal{H}, \mathcal{D}_\wedge) = \underset{\mathbf{x} \sim \mathcal{D}_\wedge}{\mathbb{E}} \left[ \frac{1}{T} \underset{\boldsymbol{\epsilon}}{\mathbb{E}} \left[ \sup_{h \in \mathcal{H}} \sum_{j=1}^T \epsilon_j \ell(h(\mathbf{x}_t(\boldsymbol{\epsilon})), h_\rho(\mathbf{x}_t(\boldsymbol{\epsilon}))) \right] \right]$$

$$= \underset{\mathbf{x} \sim \Pi}{\mathbb{E}} \left[ \frac{1}{T} \underset{\boldsymbol{\epsilon}}{\mathbb{E}} \left[ \sup_{h \in \mathcal{H}} \sum_{j=1}^T \epsilon_j \ell(h(\mathbf{x}_t(\boldsymbol{\epsilon})), h_\rho(\mathbf{x}_t(\boldsymbol{\epsilon}))) \right] \right]$$

$$= \underset{\mathbf{x} \sim \Pi}{\mathbb{E}} \left[ \frac{1}{T} \underset{\boldsymbol{\epsilon}}{\mathbb{E}} \left[ \chi_A \sup_{h \in \mathcal{H}} \sum_{j=1}^T \epsilon_j \ell(h(\mathbf{x}_t(\boldsymbol{\epsilon})), h_\rho(\mathbf{x}_t(\boldsymbol{\epsilon}))) \right] \right]$$

$$+ \underset{\mathbf{x} \sim \Pi}{\mathbb{E}} \left[ \frac{1}{T} \underset{\boldsymbol{\epsilon}}{\mathbb{E}} \left[ \chi_{A^c} \sup_{h \in \mathcal{H}} \sum_{j=1}^T \epsilon_j \ell(h(\mathbf{x}_t(\boldsymbol{\epsilon})), h_\rho(\mathbf{x}_t(\boldsymbol{\epsilon}))) \right] \right]$$

$$\leq \underset{\mathbf{x} \sim \Pi}{\mathbb{E}} \left[ \frac{1}{T} \underset{\boldsymbol{\epsilon}}{\mathbb{E}} \left[ \chi_A \sup_{h \in \mathcal{H}} \sum_{j=1}^T \epsilon_j \ell(h(\mathbf{x}_t(\boldsymbol{\epsilon})), h_\rho(\mathbf{x}_t(\boldsymbol{\epsilon}))) \right] \right] + T^2 e^{-\sigma k}$$

$$\leq \underset{\mathbf{x} \sim \Pi}{\mathbb{E}} \left[ \frac{1}{T} \underset{\boldsymbol{\epsilon}}{\mathbb{E}} \left[ \chi_A \sup_{h \in \mathcal{H}} \sum_{j=1}^T \epsilon_j \ell(h(\mathbf{x}_t(\boldsymbol{\epsilon})), h_\rho(\mathbf{x}_t(\boldsymbol{\epsilon}))) \right.\right.$$
$$\left.\left. + \sum_{j: Z_t^j \neq \mathbf{x}_t(\boldsymbol{\epsilon})} \sum_{t=1}^T \underset{\epsilon_{jt}}{\mathbb{E}} \ \epsilon_{jt} \ell(h(Z_t^j), h_\rho(Z_t^j)) \right] \right] + T^2 e^{-\sigma k}$$

$$\leq \underset{\mathbf{x} \sim \Pi}{\mathbb{E}} \left[ \frac{1}{T} \underset{\boldsymbol{\epsilon}}{\mathbb{E}} \left[ \chi_A \sup_{h \in \mathcal{H}} \sum_{j=1}^T \epsilon_j \ell(h(\mathbf{x}_t(\boldsymbol{\epsilon})), h_\rho(\mathbf{x}_t(\boldsymbol{\epsilon}))) \right.\right.$$
$$\left.\left. + \underset{\epsilon_{jt}}{\mathbb{E}} \sum_{j: Z_t^j \neq \mathbf{x}_t(\boldsymbol{\epsilon})} \sum_{t=1}^T \epsilon_{jt} \ell(h(Z_t^j), h_\rho(Z_t^j)) \right] \right] + T^2 e^{-\sigma k}$$

$$\leq \underset{\mathcal{D}}{\mathbb{E}} \left[ \frac{1}{T} \underset{\epsilon_{jt}}{\mathbb{E}} \left[ \sup_{h \in \mathcal{H}} \sum_{j=1}^k \sum_{t=1}^T \epsilon_{jt} \ell(h(Z_t^j), h_\rho(Z_t^j)) \right] \right] + T^2 e^{-\sigma k}$$

$$\leq T^2 e^{-\sigma k} + \mathcal{R}_{kT}(\ell \circ \mathcal{H}). \tag{38}$$

$\square$

Here the first inequality comes from the fact that coupling constructed in the previous section bounds the probability of $\mathbf{x}_t(\boldsymbol{\epsilon}) \notin \{Z_t^j\}_{j=1}^k$ at least for one value of $t$. The second inequality follows from the fact that $\sigma_t$ has a zero mean while the third one follows Jensen's inequality.

*Proof of Theorem 5.3.* As per Equation (35) $\mathcal{V}_T$ is upper bounded by the sequential Rademacher complexity $\mathfrak{R}_T(\mathcal{H})$. Therefore it suffices to establish an upper bound on the latter in the smoothed setting considered here. Equation (38) bounds the distribution dependent sequential Rademacher complexity with the standard notion of Rademacher complexity which assumes independent samples. Assuming the loss function to be $L$-Lipschitz, the quantity $\mathcal{R}_{kT}(\ell \circ \mathcal{H})$ can be upper bounded as:

$$\mathcal{R}_{kT}(\ell \circ \mathcal{H}) \leq L \mathcal{R}_{kT}(\mathcal{H}). \tag{39}$$

The sequential Rademacher complexity can be bounded above by the sequential fat-shattering dimension. Likewise, one can establish upper bounds on $\mathcal{R}_{kT}(\mathcal{H})$:

$$\mathcal{R}_{kT}(\mathcal{H}) \leq \inf_{\alpha>0} \left\{ 4\alpha kT + 12\sqrt{kT} \int_\alpha^1 \sqrt{K\mathrm{fat}_{c\delta}(\mathcal{H}) \log \frac{2}{\delta}} d\delta \right\}, \tag{40}$$

where $K$ and $c$ are constants. Note here that $\mathrm{fat}_\delta(\mathcal{H})$ unlike its sequential counterparts assume independent data. Therefore one can recover the definition of $\mathrm{fat}_\delta(\mathcal{H})$ from their sequential version by replacing the $\mathcal{X}$ and $\mathbb{R}$-valued tree by the sets $\mathcal{X}$ and $\mathbb{R}$. Combining Equations (38) to (40), and setting $k = \frac{2\log T}{\sigma}$ we get:

$$\mathfrak{R}_T(\ell \circ \mathcal{H}, \mathcal{D}_\wedge) \leq 1 + L \inf_{\alpha>0} \left\{ \frac{8\alpha T \log T}{\sigma} + 12\sqrt{\frac{2T \log T}{\sigma}} \int_\alpha^1 \sqrt{K\mathrm{fat}_{c\delta}(\mathcal{H}) \log \frac{2}{\delta}} d\delta \right\}. \tag{41}$$

Now using the relation that $\mathrm{fat}_\delta(\mathcal{H}_n) = O(n/\delta^2)$ when $\mathcal{H}_n = \{\mathrm{Tr}_\omega, \omega \in \mathcal{C}_n\}$ and setting $K = c = 1$ we get:

$$\mathfrak{R}_T(\ell \circ \mathcal{H}_n, \mathcal{D}_\wedge) \leq 1 + L \inf_{\alpha>0} \left\{ \frac{8\alpha T \log T}{\sigma} + 12\sqrt{\frac{2nT \log T}{\sigma}} \int_\alpha^1 \sqrt{\frac{1}{\delta^2} \log \frac{2}{\delta}} d\delta \right\}. \tag{42}$$

Finally to eliminate the infimum in Equation (42) we recall that $\alpha \in [0,1]$ and therefore for any function $f$ on $\alpha$ we get $\inf_\alpha f(\alpha) \leq f(\alpha = \alpha^\star)$; $\alpha^\star \in [0,1]$. Thus setting $\alpha = \sqrt{\frac{n\sigma}{T \log T}}$, we get:

$$\mathfrak{R}_T(\ell \circ \mathcal{H}_n, \mathcal{D}_\wedge) = O\left( \sqrt{\frac{nT \log T}{\sigma}} \right). \tag{43}$$

$\square$

