# OpenReview forum: "Online Learning of Pure States is as Hard as Mixed States"
_NeurIPS.cc/2025/Conference — NeurIPS 2025 poster_

### Official Review · Reviewer_opiU · 2025-06-30

**Clarity:** 4
**Significance:** 2
**Originality:** 2
**Rating:** 4
**Confidence:** 3

**Summary:**

This paper focuses on the problem of online learning for pure and mixed quantum states. The authors establish the equivalence between these two settings by first proving a lower bound on the sfat dimension and then demonstrating the equivalence for $L_1$-loss regret using Rakhlin’s characterization.
Furthermore, they extend the results to a more practical setting called $\epsilon$-realizable and discuss the concept of smoothness to better understand the effects of adversariality.

**Questions:**

1.	Is it correct that your work closes the sfat dimension of pure states? What would be required to close the offline version, the $\delta$-fat-shattering dimension?

**Ethical Concerns:**

["NO or VERY MINOR ethics concerns only"]

**Final Justification:**

The authors explained the technical contributions in their rebuttal, primarily focusing on two points: 1) While the final construction may seem simple, arriving at it required non-trivial insight. 2) Applying matrix completion methods to quantum/online learning, offering new intuition about instance hardness. I find both points reasonable, yet remain uncertain whether they sufficiently elevate the paper's overall contribution. Therefore, I maintain my original score, though I would support acceptance if space permits.

**Limitations:**

Yes

**Paper Formatting Concerns:**

No comments

**Quality:**

3

**Strengths And Weaknesses:**

Strengths:

1.	This paper provides rigorous proofs for every result it presents.

2.	The illustration of the sfat-tree construction makes the concept easy to understand.

3.	Smoothness is quite an interesting topic to discuss, as the adversariality of nature might not always be so extreme.

Weaknesses:

1.	While the contribution is solid, the proof of the lower bound by constructing an sfat-tree is straightforward and somewhat unsurprising. It appears to lack technical novelty.

---

> ### Author Rebuttal · Authors · 2025-07-30
>
> We thank the reviewer for their constructive feedback, and we appreciate their recognition of the rigor and clarity of our presentation.
>
> **While the contribution is solid, the proof of the lower bound by constructing an sfat-tree is straightforward and somewhat unsurprising. It appears to lack technical novelty.**
>
> Although the final tree construction may appear straightforward in retrospect, we would argue that arriving at this construction was highly non-trivial. Prior work [1] approached related lower bounds using information-theoretic tools such as random access codes and newly introduced complexity notions. In contrast, our proof is stronger in the sense that it is conceptually simpler, fully constructive, and applies uniformly to both pure and mixed quantum states, and potentially other classes of quantum states.
>
> More importantly, it explicitly identifies which instances of pure states are hardest to learn in the online adversarial setting. Those states, used in our tree construction, are characterized by a high concentration of probability mass on a small number of basis states, with very low probability spread over the remaining basis states. In addition, our utilization of matrix completion techniques provide the intuition that other types of pure states indeed won't typically be as hard to learn.
>
> Our proof technique therefore allows us to identify which instances of a hypothesis class are hardest to learn. We believe that this identification of the hardest instances within a given hypothesis class may have many applications. For example, in classical learning theory, our constructions may inform the design of secure or privacy-preserving systems. The goal is to protect a model accessed as a black-box against extraction through querying. In such settings, our techniques could offer a theoretical foundation for identifying hypothesis classes that are provably hard to reconstruct.
>
> To the best of our knowledge, we are the first to introduce matrix completion techniques to the quantum and online learning settings. Moreover, we view the simplicity of our proof, together with its potential to extend to other relevant settings, as a technical novelty in its own right.
>
>
> **Is it correct that your work closes the sfat dimension of pure states? What would be required to close the offline version, the $\delta$-fat-shattering dimension?**
>
> Yes, our work closes the asymptotic characterization of the sequential fat-shattering dimension for pure quantum states in the online setting. Specifically, we establish that pure and mixed states exhibit nearly the same sequential complexity, thereby resolving the gap that remained in prior works.
>
> However, closing the gap for the offline version would require a similar tight bound on $\delta$-fat-shattering dimension. From our construction, one might infer a lower bound of $\Omega(1/\delta^2)$. But whether the optimal bound of $\Omega(n/\delta^2)$—known to hold for mixed states—also extends to pure states is unresolved. This remains an important and intriguing direction for future work.
>
> We thank the reviewer again for their time and thoughtful engagement with our work.
>
> **References**
>
> [1] Aaronson, S., Chen, X., Hazan, E., Kale, S., and Nayak, A. Online learning of quantum states. Advances in neural information processing systems, 31, 2018.

---

> > ### Author Response · Authors · 2025-08-06
> >
> > We hope that our answers clarified the technical novelty of our work. We remain available to provide any further clarification before the discussion period concludes.
> >
> > We thank the reviewer for their time and feedback.

---

### Official Review · Reviewer_Yd1X · 2025-06-30

**Clarity:** 3
**Significance:** 3
**Originality:** 3
**Rating:** 5
**Confidence:** 4

**Summary:**

Quantum state tomography seeks to reconstruct an unknown quantum state $\rho$ from measurements. In standard (offline) settings, learning \emph{pure} states is substantially easier than learning general \emph{mixed} states, with the sample complexity for pure states scaling as $\widetilde{\Theta}(N^2)$ versus $\widetilde{\Theta}(N^4)$ for mixed states, where $N = 2^n$ for $n$ qubits. However, these results do not account for adversarial or sequential measurement scenarios.


In the online learning framework, learning proceeds as a repeated game over $T$ rounds. In each round $t$, the adversary selects a measurement $E_t$, and the learner predicts $\mathrm{Tr}(E_t \omega_t)$ for some hypothesis state $\omega_t$. The learner then incurs a loss (e.g., $\ell_1$ or $\ell_2$) with respect to the true value $\mathrm{Tr}(E_t \rho)$. The performance is measured by the \emph{regret} denoted $V_T(\mathcal{C}_n)$
where $\mathcal{C}_n$ is the set of $n$-qubit quantum states.

Their main results are as follows:
 \textbf{No separation in online learning:} The paper proves that, in the online learning framework, \emph{learning pure states is as hard as learning mixed states}. Both classes have essentially the same \emph{sequential fat-shattering dimension}, leading to identical regret scaling.

 \textbf{Regret scaling:} For both pure and mixed states, the minimax regret scales as
$$
    V_T = \Theta(\sqrt{nT})
    $$
    where $n$ is the number of qubits and $T$ is the number of rounds.
    \item \textbf{Technical approach:} The authors provide new lower bounds on the sequential fat-shattering dimension for pure states, matching those for mixed states, by constructing information-extracting trees and using matrix completion arguments.

They have some additional results as well
 In the \(\epsilon\)-realizable setting (where the learner receives noisy feedback), the lower bound on regret remains unchanged and is independent of $\epsilon$.
 In the \emph{smoothed analysis} setting (where the adversary is restricted to be close to a fixed distribution), the regret upper bound improves to
        $$
        V_T = O\left(\sqrt{\frac{nT \log T}{\sigma}}\right)
        $$
        where $\sigma$ quantifies the degree of adversariality.

\textbf{Contrast with other settings:} In some restricted online learning scenarios (e.g., bandit feedback), pure states can be learned exponentially faster than mixed states. However, in the general online learning setting considered here, this separation vanishes.
\end{itemize}

\textbf{Implications:}

The result is counterintuitive compared to classical learning, where restricting the hypothesis class (e.g., to low-rank matrices) typically reduces complexity.
In adversarial or highly dynamic experimental settings, learning pure quantum states cannot be made significantly easier than learning mixed states, despite the much lower sample complexity for pure states in standard tomography.

**Questions:**

None.

**Ethical Concerns:**

["NO or VERY MINOR ethics concerns only"]

**Limitations:**

Mentioned above.

**Paper Formatting Concerns:**

None.

**Quality:**

3

**Strengths And Weaknesses:**

Strengths:

- I think the paper addresses a fairly natural question and makes a surprising contribution.
- I like the idea of using smoothed analysis for quantum learning (as it has shown some interesting results classical recently)
- Understanding pure state versus mixed state learning is a natural question

Weakness:
- It is not very clear to me at least from the introduction, what is the main technical challenge of their work? I do feel many of their results extend straightforward from classical techniques, but it'd be nice if they made clear what was the roadblocks in establishing these results.

- I think from an ML perspective and quantum perspective the technical novelty is a tad limited (again the authors should correct me if wrong), but i still its a nice surprising result.

- The bounds are non-optimal, did you make any efforts in tightening the upper and lower bounds? It seems like the bounds that are shown here directly follow from classical results (which are quantized in this paper).

---

> ### Author Rebuttal · Authors · 2025-07-30
>
> We are grateful to the reviewer for their insightful and constructive review, which offered a clear and comprehensive summary of our work, while recognizing its significance. In what follows, we aim to clarify its technical contributions and broader relevance.
>
> **It is not very clear to me at least from the introduction, what is the main technical challenge of their work? I do feel many of their results extend straightforward from classical techniques, but it'd be nice if they made clear what was the roadblocks in establishing these results. I think from an ML perspective and quantum perspective the technical novelty is a tad limited (again the authors should correct me if wrong), but i still its a nice surprising result.**
>
> We sincerely thank the reviewer for mentioning this. While we do not introduce new concepts in online learning theory, we provide new proof strategies inspired from adjacent fields such as matrix completion. Based on this, we are able to make our proof both constructive and broadly interpretable. We consider this simplicity as a form of technical novelty.
>
> Prior work [1] approached the problem using advanced tools from information theory (e.g., random access codes) and introduced a new complexity notion (the fine-shattering dimension). By contrast, our proof is fully constructive, conceptually simpler, and yields a stronger result for online learning: it extends to pure states and explicitly identifies which instances of pure states are hardest to learn in the adversarial online setting. These extremal pure states (those used in our tree construction) are characterized by a high concentration of probability mass on a small number of basis states, with very low probability spread over the remaining basis states.
>
> We believe that this insight into which instances of a given hypothesis class are hardest to learn has potential implications beyond the quantum setting. For instance, in classical learning theory, such constructions could inform the design of secure or privacy-preserving systems, where a model is deployed as a black-box predictor and must be protected from extraction via adaptive queries. Our techniques could thus provide a theoretical foundation for identifying classes of hypotheses that are provably hard to recover in such scenarios.
>
> **The bounds are non-optimal, did you make any efforts in tightening the upper and lower bounds?**
>
> The lower bound we establish on the sequential fat-shattering dimension is $\text{sfat}_\delta(\mathcal H, \mathcal{X}) = \Omega(n/\delta^\eta)$ for all $\eta < 2$ asymptotically matches the known upper bound of $O(n/\delta^2)$. We have explored several avenues for tightening this lower bound further, but did not find a sharper construction within our framework. That said, the gap does not affect the final regret bound, which remains asymptotically tight—which is what we are concerned with in the online learning setting, where performance is evaluated in terms of rate of convergence. In this sense, the sequential complexity bound we provide is sufficient for our main results.
>
> We thank the reviewer again for their time, careful reading, and constructive engagement with the work.
>
> **References**
>
> [1] Aaronson, S., Chen, X., Hazan, E., Kale, S., and Nayak, A. Online learning of quantum states. Advances in neural information processing systems, 31, 2018.

---

### Official Review · Reviewer_NRAe · 2025-07-02

**Clarity:** 3
**Significance:** 3
**Originality:** 3
**Rating:** 5
**Confidence:** 4

**Summary:**

This work explores quantum state learning with online learning theory. The authors prove that pure and mixed states share nearly the same sequential fat-shattering dimension, which is a key measure in online learning that dictates the performance of a learning algorithm. As a result, the regret scales identically for both types of states. This is a significant finding because, in standard settings, pure states are known to be much easier to learn than mixed states. The paper also extends its findings to more practical scenarios, including the case where measurements are not perfectly accurate, and smoothed analysis.

**Questions:**

1. The paper's central result is that the separation in learning difficulty between pure and mixed states vanishes in the online setting. What are the specific properties of the adversarial online framework, as opposed to standard tomography and other quantum online learning works, that cause this equivalence in hardness?
2. What is the main advantage of your new proof strategy from previous results? Is it possible to adapt this strategy in the classical learning theory?
3. For the smoothed analysis, what would be a practical, experimental situation where the adversary's measurements correspond to $sigma$-smooth rather than completely adversarial or completely random?

**Ethical Concerns:**

["NO or VERY MINOR ethics concerns only"]

**Limitations:**

The limitations of this work are included in the Strengths and Weaknesses.

**Paper Formatting Concerns:**

I have no paper formatting concerns

**Quality:**

3

**Strengths And Weaknesses:**

Strengths:
1. The central claim—that online learning of pure quantum states is as hard as learning mixed states—is highly impactful. It challenges the common intuition derived from standard tomography, where a clear separation in learning complexity exists. This can inspire the following research, including the classical learning theory.
2. The authors introduce a new proof for the sequential fat-shattering dimension by the matrix completion. It is interesting to see new techniques coming out from the intersection between quantum and learning.
3. The authors also introduce several extensions, including a realizable setting, smoothed analysis, and L2 loss. These additions make this work comprehensive.

Weakness:
1. As also mentioned in the conclusion and limitation section, it is not clear whether the conclusion for the online learning setting is still true for the offline setting (difference between sequential fat-shattering dimension and fat-shattering dimension). If true, this can lead to an even stronger result.
2. The main result focuses on the L1 loss case. Authors also have some discussions on the L2 loss case, but not perfect. It feels a lack of explanation why L1 loss is important, or what are the technical limitations that authors can not explore other loss.
3. For the smoothed analysis part, it is not clear when such a smooth distribution could be possible in practice and why this generalization is important for quantum.

---

> ### Author Rebuttal · Authors · 2025-07-30
>
> We thank the reviewer for their detailed review. We are especially glad that they found our main result impactful and appreciated the novelty of our techniques.
>
> **The main result focuses on the L1 loss case. Authors also have some discussions on the L2 loss case, but not perfect. It feels a lack of explanation why L1 loss is important, or what are the technical limitations that authors can not explore other loss.**
>
> We thank the reviewer for raising this point. The focus on the $ L_1 $ loss is closely tied to the fact that Rademacher complexity is naturally aligned with $ L_1 $-loss. In particular, standard symmetrization techniques and contraction inequalities used in our regret analysis are sharpest when applied to $ L_1 $ loss.
>
> Extending our results to other losses, such as $ L_2 $, requires either extra constraints—such as the one we have in our derivations—or adapting existing complexity measures to take into account the curvature of the loss function. The latter falls outside the scope of our current analysis.
>
> Moreover, we note that the $ L_1 $ loss is commonly used in quantum state estimation tasks due to its operational relevance in distinguishing quantum states [1,2,3], which further justifies our focus.
>
> **The paper's central result is that the separation in learning difficulty between pure and mixed states vanishes in the online setting. What are the specific properties of the adversarial online framework, as opposed to standard tomography and other quantum online learning works, that cause this equivalence in hardness?**
>
> A key feature of the adversarial online setting is that the sequence of measurements is chosen adaptively, based on the information available to the learner at the time. This adversarial adaptivity prevents the learner from exploiting the low-rank (rank 1) structure of pure states as is the case in the stochastic or batch settings—where the measurements are entirely chosen by the learner. In this sense, although pure states are intrinsically simpler than mixed states, the online adversarial framework highlights that simplicity alone is not sufficient: a learning strategy must actively capitalize on this simplicity. However, this strategy cannot be implemented in the online adversarial setting where the measurements can be chosen adversarially.
>
> **What is the main advantage of your new proof strategy from previous results? Is it possible to adapt this strategy in the classical learning theory?**
>
> Our response to Reviewer~Yd1X provides a detailed account, but we briefly highlight the key idea here. Unlike prior approaches that rely on newly defined complexity measures or information-theoretic tools, our proof strategy is conceptually simpler and fully constructive. This not only extends existing regret lower bounds to the setting of pure quantum states, but also identifies the specific class of pure states that are the hardest to learn in the online adversarial setting.
>
> We believe that this characterization of hard instances within a hypothesis class has potential relevance beyond the quantum setting. In particular, similar constructions may be useful in classical learning contexts involving black-box predictors subjected to adaptive querying. Such scenarios arise in the design of secure or privacy-preserving systems, where identifying hypothesis classes that are provably resistant to extraction is a central challenge.
>
> **For the smoothed analysis, what would be a practical, experimental situation where the adversary's measurements correspond to sigma-smooth rather than completely adversarial or completely random?**
>
> Thanks for mentioning this. We feel that smoothed analysis may be useful for considering the noise in quantum measurements. In practice, quantum devices rarely implement measurements with perfect precision. Indeed, small perturbations (typically evolving through time) in control parameters can cause the executed measurement to deviate slightly from the intended one [4]. From a learning theory perspective, this can be modelled by the measurements being sampled from a $\sigma$-smooth distribution centred around the intended distribution. Thus, smoothed analysis captures the realistic scenario where measurement choices are subject to noise.
>
>
> We thank the reviewer once again for their time, careful reading, and thoughtful feedback.
>
> **References**
>
> [1] Aaronson, S. The learnability of quantum states. Proceedings of the Royal Society A: Mathematical, Physical and Engineering Sciences, 463(2088):3089–3114, September 2007.
>
> [2] Aaronson, S. Shadow tomography of quantum states. In Proceedings of the 50th Annual ACM SIGACT Symposium on Theory of Computing, STOC 2018.
>
> [3] Chen, X., Hazan, E., Li, T., Lu, Z., Wang, X., and Yang, R. Adaptive online learning of quantum states. Quantum, 8:1471, September 2024.
>
> [4] Leibfried, D., Blatt, R., Monroe, C., \& Wineland, D. Quantum dynamics of single trapped ions. Reviews of Modern Physics, 75(1), 281, 2003.

---

> > ### Comment · Reviewer_NRAe · 2025-08-05
> >
> > I thank the authors for the detailed response.

---

### Decision · Program_Chairs · 2025-09-17

**Decision:**

Accept (poster)

**Comment:**

(a) Summary: This paper studies the complexity of learning pure and mixed quantum states in the online setting. In the standard tomography setting, learning pure states is easier than learning mixed states. The main result of this paper shows that in contrast, pure states are as hard to learn as mixed states in the online tomography setting. This paper also extends the online setting to the smoothed and $\epsilon$-realizable settings.

(b) Strengths:
- Solid result :The paper addresses a well-motivated and natural question, and its main result is a nice message that will be useful to the broader community.
- Technical soundness: The technical results are rigorous. In addition, a few reviewers appreciated bringing smoothed analysis to online learning of quantum states.
- Clarity: The paper is very well-organized and clearly written.

(c) Weaknesses of the Paper and Missing Elements:
- Technical novelty: A recurring concern (Reviewers Yd1X, opiU) was the lack of technical novelty -- the proof techniques seem to use standard tools in a straightforward manner.
- Limitations: The main result only applies to the online setting, and it remains an open question whether the equivalence in hardness extends to the offline case. The analysis also focuses predominantly on the L1​ loss, with less discussion on other loss functions such as the L2 loss.

(d) Rationale: This paper makes a nice conceptual contribution in showing that learning pure states is as hard as learning mixed states in the online setting, contrary to known results in the standard setting. While there are some concerns about technical novelty, the consensus among reviewers is positive, and the work is technically solid and well-executed.

(e) Summary of Rebuttal and Discussion: After the rebuttal and discussions, all reviewers had a positive view of the paper. The major concerns were:
- Technical Novelty (Reviewers Yd1X, opiU): The main concern was that the proof techniques seemed straightforward. Authors provided detailed responses that seemed to satisfy reviewer Yd1X, while to a lesser extent opiU.
- Limitations (Reviewer NRAe): The reviewer mentioned that the results only apply to the online setting and the L1 loss. The authors argued that the L1 loss is commonly used in quantum tomography, and adapting the results to the L2 case requires additional analysis that is out of the scope of the paper.